# DEEP LEARNING WITH LEARNABLE PRODUCT-STRUCTURED ACTIVATIONS

**Saanjali Maharaj**
University of Toronto
saanjali.maharaj@mail.utoronto.ca

**Prasanth B. Nair**
University of Toronto
prasanth.nair@utoronto.ca

## ABSTRACT

Modern neural architectures are fundamentally constrained by their reliance on fixed activation functions, limiting their ability to adapt representations to task-specific structure and efficiently capture high-order interactions. We introduce deep low-rank separated neural networks (LRNNs), a novel architecture generalizing MLPs that achieves enhanced expressivity by learning adaptive, factorized activation functions. LRNNs generalize the core principles underpinning continuous low-rank function decomposition to the setting of deep learning, constructing complex, high-dimensional neuron activations through a multiplicative composition of simpler, learnable univariate transformations. This product structure inherently captures multiplicative interactions and allows each LRNN neuron to learn highly flexible, data-dependent activation functions. We provide a detailed theoretical analysis that establishes the universal approximation property of LRNNs and their ability to mitigate the curse of dimensionality for functions with low-rank structure. Moreover, the learnable product-structured activations enable LRNNs to adaptively control their spectral bias, which is crucial for signal representation tasks. These theoretical insights are validated through extensive experiments where LRNNs achieve state-of-the-art performance across diverse domains including image and audio representation, numerical solution of PDEs, sparse-view CT reconstruction, and supervised learning tasks. Our results demonstrate that LRNNs provide a powerful and versatile building block with a distinct inductive bias for learning compact yet expressive representations.

## 1 INTRODUCTION

Neural networks equipped with simple activation functions like ReLU, Tanh, and Sigmoid have achieved remarkable success across a multitude of domains. While their simplicity is a strength, with expressivity achieved through a deep composition of layers, these standard activations possess inherent limitations. For instance, it is well known that the spectral bias of activations such as ReLU can hinder the representation of high-frequency details in signals (Rahaman et al., 2019). This has spurred a long line of research into alternative activation functions with enhanced expressivity and improved optimization properties such as convergence and gradient propagation. Early efforts led to activations such as Maxout (Goodfellow et al., 2013), which learns a piecewise linear function, while others such as Leaky ReLU (Maas et al., 2013), PReLU (He et al., 2015), Swish/SiLU (Ramachandran et al., 2018), E-Swish (Alcaide, 2018) and GELU (Hendrycks & Gimpel, 2023) often focus on improved gradient flow or adaptive non-linearities. Kunc & Kléma (2024) provide a comprehensive survey of activation functions in deep learning.

A significant leap in expressivity, particularly for representing complex, continuous signals, emerged with the advent of implicit neural representations (INRs). The effectiveness of INRs stems from specialized activation functions designed to capture fine details and high frequencies, often providing better results than standard activations combined with positional encodings (PE) (Mildenhall et al., 2020). Pioneering work by Sitzmann et al. (2020) introduced neural networks with sinusoidal activations (SIREN), which achieves remarkable fidelity in representing images, 3D shapes, and numerical solutions of partial differential equations (PDEs). Subsequent developments include Gaussian functions (Ramasinghe & Lucey, 2022), wavelet representations (WIRE) (Saragadam et al., 2023), semi-periodic damped activations (SPDER) (Shah & Sitawarin, 2024), hyperbolic oscillation func-

tions (HOSC) (Serrano et al., 2024), sinc activations (Saratchandran et al., 2024), and FINER (Liu et al., 2024); see Essakine et al. (2025) for a recent review. This body of work has demonstrated that the choice and design of the non-linearity are crucial for high-fidelity signal representation.

While tailored activation functions are powerful, they are often designed to capture specific signal properties such as periodicity and multi-scale behavior. This motivates the search for architectures that can *learn* highly expressive and adaptive non-linearities, while maintaining computational efficiency and optimization stability. A recent development in this direction is Kolmogorov-Arnold Networks (KANs) proposed by Liu et al. (2025), which incorporate learnable activation functions on edges rather than using fixed activations at nodes. While KANs offer increased expressivity, they require significantly longer training times and can suffer from optimization instability with larger grid sizes. The ActNet architecture (Ferreira Guilhoto & Perdikaris, 2025) which leverages Laczovich's variant of Kolmogorov's superposition theorem has shown promise in addressing these challenges.

In the present work, we introduce deep low-rank separated neural networks (LRNNs), a novel architecture whose expressivity stems from a different principle: a multiplicative composition of learnable, univariate functions. LRNNs are inspired by work on low-rank separated representations, originally proposed for approximating multivariate functions as sums of products of univariate basis functions (Beylkin et al., 2009; Audouze & Nair, 2019). LRNNs generalize this idea to create a new class of deep neural network architectures where the neurons are equipped with learnable product-structured activations. This structure inherently captures multiplicative interactions and allows each *neuron* to independently learn a highly flexible activation function, adapting its non-linearity to the learning task, while maintaining computational efficiency.

The notion of low-rank function decomposition that our work builds upon has deep roots in tensor algebra (Kolda & Bader, 2009). Low-rank decompositions of model weights have been successfully applied to model compression (Novikov et al., 2015; Lebedev et al., 2015) and fine-tuning large language models (LLMs) (Hu et al., 2022). However, our focus with LRNNs is distinct: rather than using low-rank decompositions for compression, we leverage the multiplicative structure of low-rank function decompositions to enhance expressivity. More specifically, LRNNs utilize adaptive product-structured activations to efficiently capture high-order interactions in contrast to standard neurons that compose features additively. Our main contributions are:

- We introduce the LRNN architecture, a generalization of MLPs where each neuron's activation is a product of learnable univariate functions applied to projected inputs, enabling highly adaptive and expressive non-linearities beyond fixed scalar activations.

- We provide detailed theoretical analysis establishing universal approximation, the ability of LRNNs to overcome the curse of dimensionality for functions with decaying functional ANOVA structure, and insights into how LRNNs can adaptively control their spectral bias.

- We demonstrate that the unique theoretical advantages of LRNNs translate into practical impact across diverse domains:

  - *Image representation*: LRNNs achieve 100% success at a high-fidelity 40 dB target across 1,000 ImageNet images, a regime where the SPDER and SIREN baselines frequently fail.
  - *Audio representation*: 3-11x lower MSE on audio tasks with superior spectral fidelity.
  - *Numerical solution of PDEs*: LRNNs achieve 8x parameter reduction compared to SIREN and 100-1000x lower error than KANs on a PDE benchmark.
  - *Sparse-view CT reconstruction*: On this benchmark, LRNNs provide artifact-free reconstruction and superior performance with a small number of projections (50-100), a critical factor for reducing patient radiation exposure.

## 2 LOW-RANK DECOMPOSITIONS IN LEARNING

Low-rank tensor decompositions provide a powerful framework for mitigating the curse of dimensionality by representing high-dimensional tensors through interactions among their dimensions, often as products of low-dimensional tensors. The origins of this topic can be traced to work by Hitchcock (1927) on decomposing a tensor into a sum of rank-one tensors, which was later refined by Cattell (1944) with parallel proportional and multi-axis analysis. The most widely used approach is arguably the canonical polyadic (CP) decomposition (Carroll & Chang, 1970; Harshman, 1970).

The Tucker decomposition (Tucker, 1966) provides a more general family of decompositions that includes the CP decomposition as a special case. The tensor train (TT) decomposition (Oseledets, 2011) is a well-studied approach that combines the advantages of CP and Tucker decompositions. It is worth noting that since the Eckart-Young-Mirsky theorem only holds for matrices, there is no unique approach for generalizing the notion of singular value decomposition to higher-order tensors (Kolda & Bader, 2009).

Low-rank tensor decompositions are now ubiquitous in many fields (Kolda & Bader, 2009), and applications include scientific computing (Dolgov et al., 2021), dimensionality reduction (Papalexakis et al., 2015; Shashua & Levin, 2001; Acar et al., 2006), compression of deep learning models (Novikov et al., 2015), reducing memory footprint when fine-tuning LLMs (Hu et al., 2022), and decomposition of LLM gradients to reduce training memory (Zhao et al., 2024).

The present work is motivated by the observation that a low-rank tensor decomposition can be interpreted as a discretization of a continuous low-rank decomposition of a multivariate function. This observation underpins the separated rank decomposition (SRD) model (Beylkin et al., 2009), which is a continuous generalization of the CP decomposition. For a $d-$dimensional function, the SRD model takes the form: $\hat{y}(\mathbf{x}) = \sum_{i=1}^{r} s_i \prod_{j=1}^{d} g_{i,j}(x_j)$, where $r$ is the separation rank, $s_i$ are normalization coefficients, and $g_{i,j}$ are approximated using a linear combination of univariate basis functions. The basis functions are typically polynomials or radial basis functions and alternating least squares is used for training (Beylkin et al., 2009; Chevreuil et al., 2015), which can result in slow convergence and ill-conditioned subproblems when the support region of the basis is disjoint from the data points. Audouze & Nair (2019) proposed a sparse SRD approach that uses $\ell_1$-regularization and a coordinate descent optimization algorithm to address these challenges.

Other models in the literature that use univariate component functions include projection pursuit regression (Friedman & Stuetzle, 1981) and neural additive models (NAMs) (Agarwal et al., 2021). Tree tensor networks (TTNs) (Shi et al., 2006; Cheng et al., 2019; Bachmayr et al., 2023; Ali & Nouy, 2023) are another class of models that use compositions of low-dimensional functions that are not restricted to be univariate. TTNs have a structure similar to an MLP equipped with a sparsity mask. Despite the growing body of theoretical work on this topic, applications have so far been restricted to simple test problems. This can be attributed to the fact that learning the optimal tree structure from data is a challenging combinatorial problem.

# 3 LRNN ARCHITECTURE

In this section, we introduce the LRNN architecture in the supervised learning setting. Let $\mathcal{D}$ denote a dataset of $N$ observations, $\mathcal{D} := \{(\mathbf{x}^{(i)}, \mathbf{y}^{(i)})\}_{i=1}^{N}$, where $\mathbf{x}^{(i)} = (x_1^{(i)}, \ldots, x_d^{(i)}) \in \mathbb{R}^d$ is an input vector with $d$ feature dimensions and $\mathbf{y}^{(i)} \in \mathcal{Y}$ denotes a noisy observation containing either $K$ regression targets (i.e., $\mathcal{Y} \subset \mathbb{R}^K$) or $K$ class labels (i.e., $\mathcal{Y} \subset \mathbb{Z}^K$). Our goal is to construct a predictive model for the regression targets or class labels using the dataset $\mathcal{D}$.

## 3.1 SHALLOW LRNNs

At the core of our approach is the LRNN neuron's ability to capture multiplicative interactions through product-structured activations. We begin with the shallow LRNN architecture, in which a single layer transforms inputs through learnable univariate functions, before extending to deeper compositions. For a dataset with $K$ outputs, the shallow LRNN takes the form

$$\widehat{\mathbf{y}}_{\text{lrnn}}(\mathbf{x}) = \sum_{\ell=1}^{r} \mathbf{s}_\ell \prod_{j=1}^{\bar{d}} (1 + \gamma\, g_j^\ell(z_j^\ell)), \ \ \mathbf{z}^\ell = \mathbf{W}^\ell \mathbf{x} + \mathbf{b}^\ell, \tag{1}$$

where $r \in \mathbb{N}$ is the separation rank, $\mathbf{s}_\ell \in \mathbb{R}^K$ are weight vectors, $g_j^\ell : \mathbb{R} \to \mathbb{R}$ denotes a univariate component function, $\gamma = \bar{d}^{-1/2}$ is a scaling factor, $\mathbf{z}^\ell \in \mathbb{R}^{\bar{d}}$, $\mathbf{W}^\ell \in \mathbb{R}^{\bar{d} \times d}$, and $\mathbf{b}^\ell \in \mathbb{R}^{\bar{d}}$.

It can be seen from (1) that LRNNs project the $d$-dimensional input to $r$ latent vectors in $\mathbb{R}^{\bar{d}}$ and produce an output using a sum-product operation. We introduce the term $(1 + \gamma\, g_j^\ell(z_j^\ell))$ to ensure automatic relevance determination (ARD) and make initialization more convenient; see Appendix A.1 for details. The scaling factor $\gamma = \bar{d}^{-1/2}$ plays a crucial role analogous to Xavier/He initialization

in standard networks (Glorot & Bengio, 2010; He et al., 2015) and the scaling used in LoRA (Hu et al., 2022). We formalize this by establishing the following result (see Appendix A.3 for proof):

**Lemma 1** (Variance-controlled initialization). *Under mild assumptions on the component functions at initialization (zero mean, finite variance), the product-structured LRNN activation $\varphi(\mathbf{z}) = \prod_{j=1}^{\bar{d}}(1 + \gamma\, g_j(z_j))$ satisfies the following bounds:*

$$(i)\ \mathrm{Var}[\varphi(\mathbf{z})]\ \leq\ e^{\sigma_g^2} - 1\ \text{ and } (ii) \sum_{k=1}^{\bar{d}} \mathrm{Var}[\partial\varphi(\mathbf{z})/\partial z_k]\ \leq\ \sigma_{g'}^2\, e^{\sigma_g^2} \tag{2}$$

*where $\sigma_g^2$ and $\sigma_{g'}^2$ denote the variance of the component functions and their first-order derivatives, respectively, at initialization.*

It follows from this result that the variance of the LRNN activation and the sum of the variances of its gradients are bounded independently of the projection width $\bar{d}$. This reveals an intrinsic mechanism for ARD: as projection width $\bar{d}$ increases, each coordinate's gradient contribution $\mathrm{Var}[\partial\varphi/\partial z_k] = \mathcal{O}(1/\bar{d})$ diminishes, while their collective impact remains constant. This ensures stable gradient flow through arbitrarily wide product structures, enabling LRNNs to learn high-dimensional yet well-conditioned representations.

The univariate LRNN component functions $g_j^\ell : \mathbb{R} \to \mathbb{R}$ can be flexibly parametrized, with each of the $r\bar{d}$ functions typically being a small MLP, enabling them to adapt to complex patterns in the data. The parameters of the component functions are learned along with the weight vectors $\mathbf{s}_\ell \in \mathbb{R}^K$ during training. The hyperparameters of a shallow LRNN are the separation rank, $r$, which controls the model's expressivity, and the dimensionality of the linear projection layer, $\bar{d}$.

**Connection to SRD:** For the special case of scalar targets (i.e., $K = 1$), if we set the projection layer to identity (i.e., $\mathbf{z}^\ell = \mathbf{x}$) and replace $(1 + \gamma\, g_j^\ell(z_j^\ell))$ with $g_j^\ell(z_j^\ell)$, we recover the SRD model of Beylkin et al. (2009). The LRNN model can hence be viewed as a generalization of CP-based function decomposition, which we will later generalize further to deeper architectures.

**Generalization of MLPs:** LRNNs generalize the familiar MLP architecture. If we set $\bar{d} = 1$ and replace $g_j^\ell$ with a standard activation function, LRNNs reduce to a standard shallow MLP. To see this generalization clearly, consider a shallow MLP with $r$ neurons in the hidden layer: $\mathbf{y}_{\mathrm{mlp}}(\mathbf{x}) = \sum_{\ell=1}^r \mathbf{v}_\ell\, \sigma(z_\ell)$, where $z_\ell = \mathbf{w}_\ell^T\mathbf{x} + b_\ell$ is a scalar projection of the input with $\mathbf{w}_\ell \in \mathbb{R}^d, \mathbf{v}_\ell \in \mathbb{R}^K$, and $b_\ell \in \mathbb{R}$ denoting the weights and biases, respectively, and $\sigma : \mathbb{R} \to \mathbb{R}$ is a standard MLP activation function. The shallow LRNN in (1) can be rewritten in the same form: $\mathbf{y}_{\mathrm{lrnn}}(\mathbf{x}) = \sum_{\ell=1}^r \mathbf{s}_\ell\, \varphi_\ell(\mathbf{z}^\ell)$, where $\varphi_\ell(\mathbf{z}^\ell) = \prod_{j=1}^{\bar{d}}(1 + \gamma\, g_j^\ell(z_j^\ell))$ is the LRNN product-structured activation function with $\mathbf{z}^\ell = \mathbf{W}^\ell\mathbf{x} + \mathbf{b}^\ell$. The key distinctions are: (i) each LRNN neuron learns its own *distinct learnable* activation function $\varphi_\ell : \mathbb{R}^{\bar{d}} \to \mathbb{R}$, whereas all MLP neurons share the same *fixed* activation $\sigma : \mathbb{R} \to \mathbb{R}$ operating on scalar projections;[1] (ii) LRNN activations achieve this vector-to-scalar mapping through multiplicative compositions, enabling efficient representation of higher-order interactions that additive architectures struggle to capture (see Section 3.3).

## 3.2   DEEP LRNNS

We now extend LRNNs to deeper architectures, enabling them to learn hierarchical representations through composed transformations. Deep LRNNs stack multiple layers, creating a sequence of maps from input to output space, i.e., $\mathbf{x}^{(0)} \to \mathbf{x}^{(1)} \to \ldots \to \mathbf{x}^{(L)} \to \widehat{\mathbf{y}}$ for a model with $L$ layers; see Figure 10 for a graphical illustration. This hierarchy progressively transforms inputs into latent representations amenable to efficient low-rank approximation, combining deep learning's compositional power with the expressivity of low-rank function decomposition.

A deep LRNN architecture with $L$ layers and $r_k$ neurons in the $k$th hidden layer for mapping a $d$-dimensional input to a $K$-dimensional output can be represented as

$$\widehat{\mathbf{y}}(\mathbf{x}) = \mathbf{S}^{\mathrm{out}}(\phi^{(L)} \circ \phi^{(L-1)} \circ \ldots \circ \phi^{(1)})(\mathbf{x}), \tag{3}$$

---

[1]Maxout networks (Goodfellow et al., 2013) are a notable exception, also using vector-to-scalar mappings but through max operations rather than products.

where $\mathbf{S}^{\text{out}} \in \mathbb{R}^{K \times r_L}$ and $\phi^{(k)} : \mathbb{R}^{r_{k-1}} \to \mathbb{R}^{r_k}$, with $r_0 = d$ and $\phi^{(0)} = \mathbf{x}$. The output of the $k$-th hidden layer can be written as $\phi^{(k)} = (\varphi_1^{(k)}(\mathbf{z}^{1,(k)}), \varphi_2^{(k)}(\mathbf{z}^{2,(k)}), \ldots, \varphi_{r_k}^{(k)}(\mathbf{z}^{r_k,(k)}))^T$, where

$$\varphi_\ell^{(k)}(\mathbf{z}^{\ell,(k)}) = \prod_{j=1}^{\bar{d}_k} (1 + \gamma\, g_j^{\ell,(k)}(z_j^{\ell,(k)})),\ \ell = 1, 2, \ldots, r_k,\ k = 1, 2, \ldots, L. \tag{4}$$

In the preceding equation $\mathbf{z}^{\ell,(k)} = \mathbf{W}^{\ell,(k)}\phi^{(k-1)} + \mathbf{b}^{\ell,(k)} \in \mathbb{R}^{\bar{d}_k}$, and $\mathbf{W}^{\ell,(k)} \in \mathbb{R}^{\bar{d}_k \times r_{k-1}}$, and $\mathbf{b}^{\ell,(k)} \in \mathbb{R}^{\bar{d}_k}$ denote the weight matrix and bias vector for the $\ell$-th neuron, respectively, $g_j^{\ell,(k)} : \mathbb{R} \to \mathbb{R}$ are learnable univariate component functions, and $\varphi_\ell^{(k)} : \mathbb{R}^{\bar{d}_k} \to \mathbb{R}$.

Each LRNN neuron $\ell$ in layer $k$ applies a product-structured activation $\varphi_\ell^{(k)}$ to a $\bar{d}_k$-dimensional projection of the previous layer's output. The matrix $\mathbf{S}^{\text{out}}$ maps $\phi^{(L)}$ to the target space. Similar to the shallow LRNN, the deep generalization also introduces distinct product-structured activation functions within and across layers that operate on distinct $\bar{d}_k$-dimensional projections.[2]

Parameter sharing can be used to reduce the parameter complexity of deep LRNNs, e.g., sharing the activation function across LRNN neurons in each layer, the term $\varphi_\ell^{(k)}$ can be rewritten as

$$\varphi_\ell^{(k)}(\mathbf{z}^{\ell,(k)}) = \prod_{j=1}^{\bar{d}_k} (1 + \gamma\, g_j^{(k)}(z_j^{\ell,(k)})),\ \text{where } \mathbf{z}^{\ell,(k)} = \mathbf{W}^{\ell,(k)}\phi^{(k-1)} + \mathbf{b}^{\ell,(k)}. \tag{5}$$

The use of shared activations for the neurons in each hidden layer reduces the number of learnable univariate component functions from $r_k \bar{d}_k$ to $\bar{d}_k$ (i.e., $g_j^{(k)}$ shared across all neurons in the $k$-th layer). Note that each hidden layer is equipped with a distinct learnable product-structured activation function. We evaluate the trade-offs of this approach in Appendix C. (Figure 11), comparing this shared activation variant against the standard flexible LRNN architecture. Our results indicate that while parameter sharing improves efficiency at lower parameter counts, distinct activations are necessary to maximize fidelity for complex high-frequency signals. Another possibility is to share the projection layer across neurons in each layer; however, we found that this approach leads to significant loss in expressivity.

**Implementation aspects:**   In our implementation, each univariate component function $g_j^{\ell,(k)}$ within the LRNN's product-structured activation is parametrized by a small shallow MLP. These component networks employ standard scalar activations: for implicit neural representation tasks, we use either SPDER activations (e.g., $\sin(x)\sqrt{|x|}$, $\sin(x)\arctan(x)$) or SIREN's sinusoidal activation ($\sin(x)$). To ensure stable learning dynamics in deep architectures, we apply LayerNorm to the output vector $\phi^{(k)}$ after each LRNN layer's product-structured computation; see Appendix C.2 for details. This normalization strategy proves crucial for consistent convergence in deeper networks. Implementation details and ablation studies are provided in Appendices B and C, respectively.

### 3.3   THEORETICAL ASPECTS

We establish fundamental theoretical properties of LRNNs that provide insight into their empirical success: universal approximation, mitigation of the curse of dimensionality for structured functions, and adaptive spectral bias control.

**Theorem 1** (Universal approximation). *If $f : [0,1]^d \to \mathbb{R}$ is a continuous function, then for every $\varepsilon > 0$, there exists an LRNN with suitably chosen separation rank $r$ such that $\max_{\mathbf{x} \in [0,1]^d} |f(\mathbf{x}) - f_{\text{lrnn}}(\mathbf{x})| \le \varepsilon$.*

This result, establishing universality analogous to that of standard MLPs, follows from the Stone-Weierstrass theorem and the fact that LRNNs can represent arbitrary polynomial expansions (see Appendix A.4 for proof). Just as the width of an MLP may grow with $1/\varepsilon$, the separation rank $r$ of an LRNN can grow arbitrarily large to capture complex functions. Thus, "universal" here does not guarantee a small $r$ unless the target function has low-rank or near-separable structure.

Beyond universal approximation, the stability of training is important in practice. It follows from Lemma 1 (Section 3.1) that variance-controlled initialization ensures variance-controlled learning for LRNNs:

---

[2]In practice, we set $\bar{d}_k$ to be the same across layers, i.e., $\bar{d}_k = \bar{d}\ \forall k$.

*The scaling factor $\gamma = \bar{d}^{-1/2}$ ensures that both forward and backward propagation remain stable regardless of projection width $\bar{d}$, with activation variance and the gradient variance sum both bounded from above by constants independent of $\bar{d}$.*

This property enables automatic relevance determination and stable optimization even for wide product structures (detailed analysis in Appendix A.3).

**Theorem 2** (Curse of dimensionality mitigation). *For functions whose ANOVA decomposition is dominated by terms involving at most $m \ll d$ variables, LRNNs achieve approximation error $\varepsilon$ with parameter complexity $\mathcal{O}(poly(d)/\varepsilon)$ rather than exponential in $d$.*

Theorem 2 shows that LRNNs can circumvent the curse of dimensionality for a class of structured functions since the parameter complexity grows only polynomially with $d$ rather than the exponential scaling typical of generic approximators. Appendix A.5 provides a more precise statement of this result with the technical assumptions and proof. The key insight is that LRNNs naturally encode sum-of-products structures matching ANOVA decompositions. Functions arising from physical systems often exhibit such decay in interaction order, making LRNNs particularly suitable for scientific computing applications. Figure 1 illustrates this in practice for a synthetic test function with product-structure; see Appendix D for details.

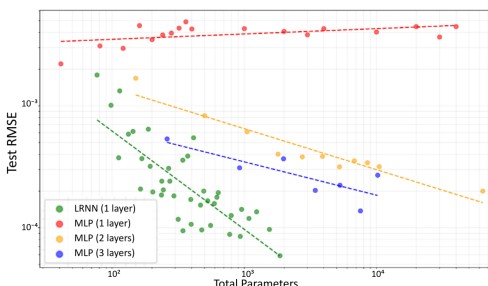

Figure 1: Test RMSE vs. parameter count for LRNN and ReLU MLP.

**Lemma 2** (Adaptive spectral bias control). *When equipped with periodic activations (e.g., SIREN, SPDER), LRNNs with $\bar{d} > 1$ generate rich frequency spectra through combinatorial frequency synthesis. A single LRNN neuron with $\bar{d}$ components generates not only the $\bar{d}$ fundamental frequencies but also all $2^{\bar{d}} - 1$ possible sum and difference combinations.*

This multiplicative frequency synthesis contrasts with MLPs' additive synthesis, where each neuron contributes a single frequency pair. Consequently, LRNNs can represent complex spectra with fewer parameters, particularly for signals with harmonic relationships or intermodulation products. This explains their superior performance on audio and image representation tasks where the ability to capture high-frequency details is important (see Appendix A.7 for proof and detailed discussion).

## 4 NUMERICAL EXPERIMENTS

We evaluate LRNNs across diverse domains, including image and audio representation, numerical solution of PDEs, and sparse-view CT reconstruction. All models were implemented in Py-Torch (Paszke et al., 2019) and trained using the Adam optimizer (Kingma & Ba, 2015) on a single NVIDIA 4090 GPU. Our implementation is publicly available at `https://github.com/dacelab/lrnn`.

Prior to presenting task-specific results, we summarize key architectural insights gained through extensive ablation studies (detailed in Appendices C–H): (i) Stability: The multiplicative structure of LRNNs alters activation statistics compared to additive networks, making LayerNorm essential for convergence (Appendix C.2); (ii) Component selection: Using periodic activations (e.g., SIREN, SPDER) within the univariate components is crucial for minimizing spectral bias in high-frequency tasks (Appendix C.3); and (iii) Robustness: LRNNs provide excellent performance in the sparse-data regime, maintaining high reconstruction fidelity (Appendix H).

**Image representation:** We conducted experiments to study how the performance, measured by peak signal-to-noise ratio (PSNR) scales with parameter count for LRNN, SPDER (Shah & Sitawarin, 2024), and MLPs. We refer to our LRNN implementation for this case as LRNN-SPDER, since we use the SPDER activation function $\sin(x)\sqrt{|x|}$ in the MLP parametrizations of the LRNN univariate component functions; see Appendix E for details.

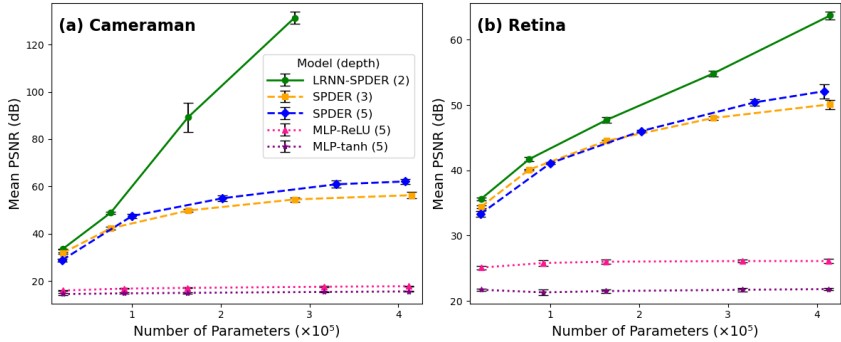

Figure 2: Scaling laws for image representation task.

As shown in Figure 2, on both the cameraman image (grayscale $256 \times 256$) and the retina image (RGB $256 \times 256$), a 2-layer LRNN-SPDER consistently outperforms deeper 3- and 5-layer SPDER models and 5-layer MLP models with both ReLU and tanh activations across all tested parameter counts. For cameraman, the performance gap widens between LRNN-SPDER and SPDER up to ∼300k parameters, suggesting that LRNN-SPDER achieves superior parameter efficiency compared to its baseline. Similarly, for retina, 2-layer LRNN-SPDER shows increasing performance gains as model size grows compared to both SPDER models. These scaling plots highlight LRNN-SPDER's expressivity and learning capacity, surpassing the corresponding benchmark on which the LRNN activations are based, even at shallower depth.

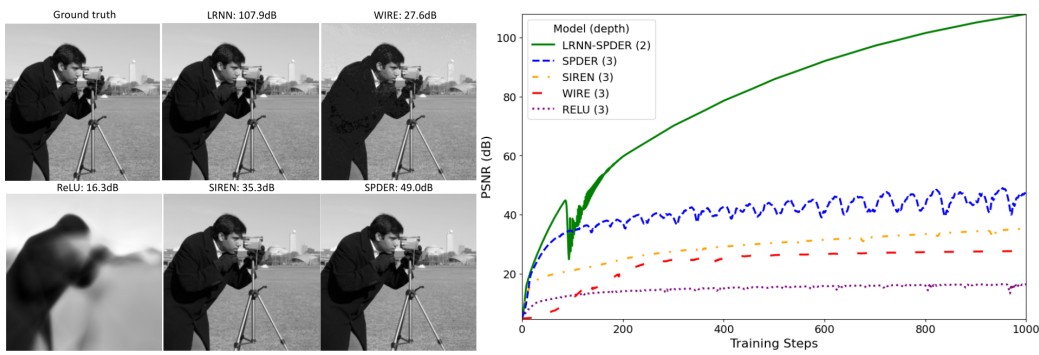

Figure 3: Cameraman image results: ground truth and reconstructed images from LRNN-SPDER, SPDER, SIREN, WIRE and ReLU (left); and PSNR convergence history over iterations (right).

We compare the performance of LRNN-SPDER to an MLP with ReLU activations, SIREN (Sitzmann et al., 2020), WIRE (Saragadam et al., 2023) and SPDER (Shah & Sitawarin, 2024) on the cameraman image. All models are chosen to have ∼197k parameters. LRNN-SPDER outperformed all the baselines, achieving a PSNR of 107.9 dB–a margin of 58.9 dB over the next best, SPDER (49.0 dB). While such extreme fidelity exceeds visual distinguishability, it confirms that LRNNs avoid the spectral saturation limiting standard architectures. Figure 3 shows the qualitative image reconstructions and PSNR convergence history for all models. The PSNR convergence plot shows LRNN-SPDER achieving the highest PSNR early on in training and continuing to climb even while the other models show signs of convergence.

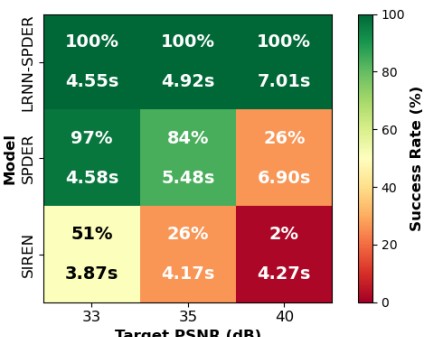

Figure 4: Average success rate and time for models to reach PSNR targets on 1000 ImageNet images.

We conducted a large-scale robustness study on 1,000 images from the ImageNet dataset, each postprocessed to $256 \times 256$. LRNN-SPDER, SIREN, and SPDER models with ∼200k parameters were trained for 1,000 epochs using three random seeds per image, totaling 3,000 runs per model. Figure 4 reports the success rate and average wall-clock time required to reach PSNR targets of 33 dB, 35 dB, and 40 dB. We observe that LRNNs consis-

tently outperform the baselines. For the challenging 40 dB target, LRNNs achieved a $100\%$ success rate, whereas SIREN and SPDER failed to reach this target in $98.2\%$ and $73.6\%$ of cases, respectively. Even at lower targets, LRNNs surpass SPDER in both success rate and time-to-solution. While SIREN has shorter runtimes, its success rate significantly drops at higher fidelity targets. See Appendix E.1 for further analysis. Figure 5 compares representative reconstructions at epoch 250. LRNN reconstructions are virtually indistinguishable from the ground truth, whereas SPDER exhibits mild degradation and SIREN produces blurrier outputs.

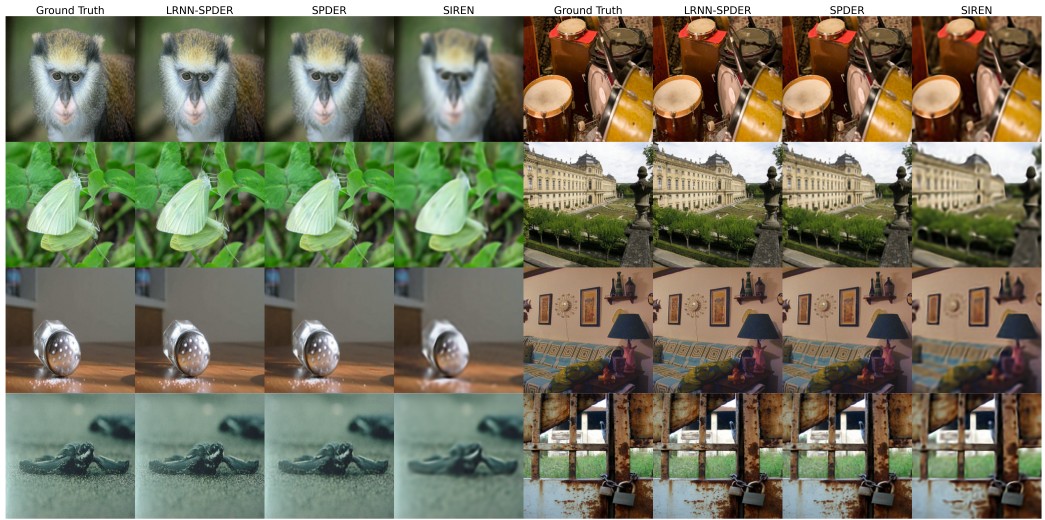

Figure 5: Qualitative comparison of ImageNet reconstructions after 250 epochs. Selected examples demonstrate that LRNNs capture fine details significantly earlier in training than baseline models.

We demonstrate the capabilities of LRNN-SPDER on images from the DIV2K dataset. We downsample the original image with a scaling factor of 4, train the model on the downsampled image, then reconstruct the image by upsampling the model output back to the full resolution. The frequency factor $\omega_0$ within the MLP component functions of our LRNN-SPDER model was chosen following de Avila Belbute-Peres & Kolter (2023). The upsampled reconstruction is compared to the ground truth in Figure 6. Additional studies on color images are presented in Appendix E.

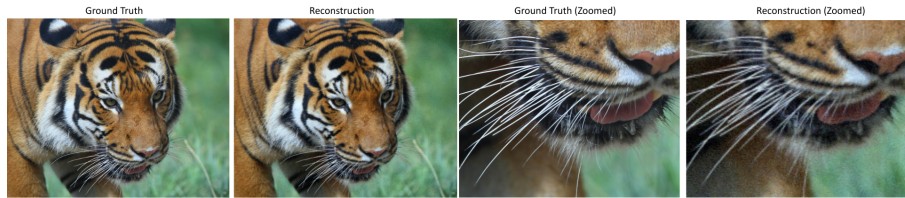

Figure 6: LRNN-SPDER upsampled image reconstruction demonstrated on DIV2K image.

**Audio Representation:** We tested LRNNs on audio signal representation using four diverse clips: instrumental classical music (bach) and male human speech (counting) as used in Sitzmann et al. (2020); reggae music with singing (reggae) from the GTZAN dataset (Tzanetakis et al., 2001); and female human speech (reading) from the LibriSpeech dataset (Panayotov et al., 2015). We use $\sin(x)\arctan(x)$ (Shah & Sitawarin, 2024) activations for the MLP-based component functions of LRNN-SPDER and compare against the baseline models, SIREN and SPDER. Table 1 presents MSE loss and frequency similarity ($\rho_{AG}$) mean values (std) over 10 runs, while Figure 7 shows time and frequency domain absolute errors and loss convergence for the bach audio. Additional results and implementation details are presented in Appendix F. The results consistently demonstrate LRNNs' superiority:LRNN-SPDER significantly outperforms all other models, including its SPDER baseline, achieving **3x–11x** lower final MSE and higher $\rho_{AG}$ across all the datasets.

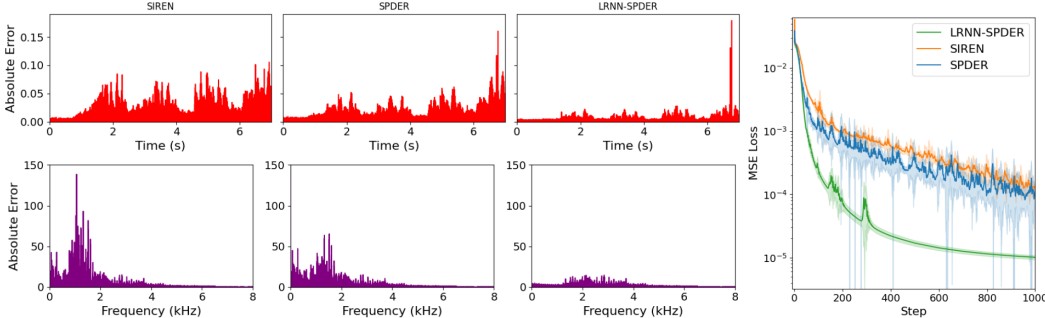

Figure 7: Absolute error in time and frequency domain and convergence of training MSE loss (mean $\pm 1\sigma$) for bach audio representation tasks for comparably sized models.

LRNN-SPDER also exhibits faster convergence compared to the baselines. The error distributions in the frequency domain show that LRNN-SPDER preserves spectral integrity essential to human perception (e.g., timbre, pitch, harmonics). This demonstrates superior generalization across both temporal and spectral representations compared to baselines.

Table 1: Comparison of MSE loss and $\rho_{AG}$ on Audio datasets. Values show mean(std) over 10 runs.

| Method | MSE Loss ($\times 10^{-4}$) | | | |
|---|---|---|---|---|
| | bach | counting | reggae | reading |
| SIREN | 1.21(0.28) | 2.77(0.56) | 21.5(6.3) | 9.98(1.57) |
| SPDER | 1.12(0.05) | 2.29(0.55) | 24.8(7.7) | 8.88(2.45) |
| LRNN-SPDER | **0.10(0.01)** | **0.72(0.03)** | **7.93(0.11)** | **1.86(0.30)** |
| | $\rho_{AG}$ (std $\times 10^{-4}$) | | | |
| SIREN | 0.9986(5) | 0.9906(15) | 0.9769(11) | 0.9193(94) |
| SPDER | 0.9988(3) | 0.9937(6) | 0.9729(10) | 0.9324(104) |
| LRNN-SPDER | **0.9999(0)** | **0.9967(2)** | **0.9860(2)** | **0.9862(31)** |

**PDE benchmark:** We evaluate LRNNs (with $\sin(x)$ activations for the MLP component functions) on the high-frequency Poisson PDE benchmark (Liu et al., 2025), comparing against SIREN, MLPs, and KANs. Figure 8 compares the mean squared error (MSE) obtained using LRNN, MLP, and SIREN with different model parameter counts when the frequency parameter is set to $n = 2$ and $n = 4$. Results for KAN1 ([100] $G = 10$) and KAN2 ([100] $G = 20$) are from Liu et al. (2025) and are displayed as horizontal lines since the parameter counts are unknown. It can be seen from the results that LRNNs exhibit exceptional performance, particularly in parameter efficiency. For instance, a 16k-parameter 2-layer LRNN achieves a significantly lower error for frequency $n = 2$ and comparable error at $n = 4$ relative to a 132k-parameter SIREN with three hidden layers (an **8x** parameter reduction).

Furthermore, a 57k-parameter LRNN reduces the error by nearly an order of magnitude compared to the 132k-parameter SIREN when $n = 4$. Compared to KANs, LRNNs achieve orders of magnitude lower error across all tested frequencies. These results underscore LRNNs' superior expressivity for complex PDE solutions with compact models; see Appendix G for additional details and results.

**CT Reconstruction:** Sparse-view Computed Tomography (CT) is vital for reducing patient radiation exposure, and INRs can reconstruct high-fidelity images from such limited data. We compared LRNN against WIRE, SIREN, Gaussian-activated networks (Gauss), and ReLU with positional encoding on a $256 \times 256$ chest CT image task (Saragadam et al., 2023), using $\sim$180k parameters for all models.

Table 2 shows that LRNN achieves the highest PSNR (29.13 dB) and SSIM (0.7455). Qualitatively, Figure 9 demonstrates LRNNs' superior reconstruction fidelity: its output is sharper and closer to the ground truth than the blurry results from SIREN and ReLU+PE. Notably, while the training loss of the LRNN converges similarly to WIRE (the next best in PSNR), the LRNN reconstruction is free from the high-frequency artifacts present in WIRE's output. This suggests that LRNNs find solutions that better correspond to perceptually accurate image features. Appendix H further details ablation studies, confirming the suitability of LRNNs for sparse-view CT. This artifact-free reconstruction from limited projections has direct clinical implications for reducing patient radiation exposure while maintaining diagnostic quality.

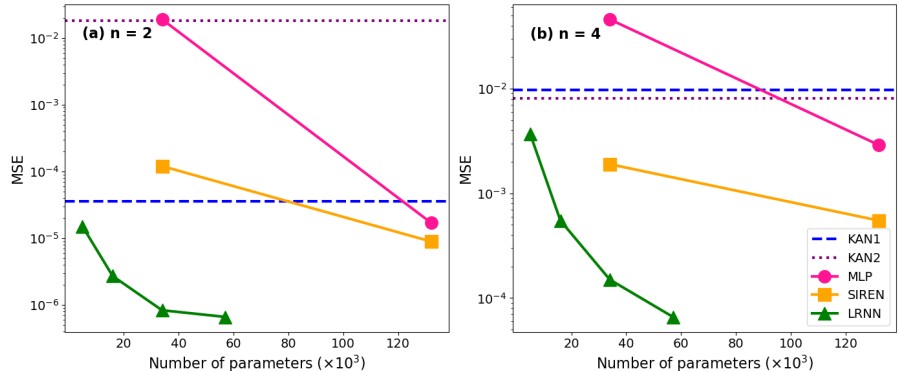

Figure 8: Results for the PDE benchmark.

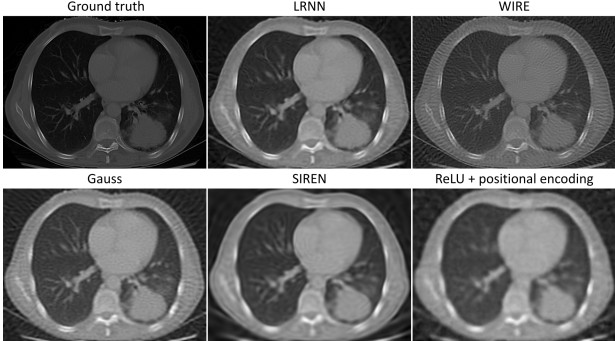

Figure 9: CT ground truth and reconstructed images.

Table 2: CT model performance comparison.

| Model | PSNR | SSIM |
|-------|------|------|
| LRNN | **29.13** | **0.7455** |
| WIRE | 28.83 | 0.6413 |
| Gauss | 27.84 | 0.6855 |
| SIREN | 27.46 | 0.6877 |
| ReLU | 26.89 | 0.6341 |

## 5 CONCLUDING REMARKS

We introduced deep low-rank separated neural networks (LRNNs), a novel architecture that generalizes MLPs, achieving enhanced expressivity through learnable product-structured activation functions. This design allows LRNN neurons to effectively capture complex high-order interactions with a compact parameterization. Our theoretical analysis established LRNNs' universal approximation capabilities, their potential to overcome the curse of dimensionality for functions with low-rank structure, and their ability to adaptively control spectral bias–crucial for signal representation tasks.

Our extensive experiments demonstrate that LRNNs hold significant potential across several domains. They set new benchmarks on a challenging PDE test-case, achieving orders of magnitude lower error with significantly fewer parameters than SIREN, MLPs, and KANs. In INR tasks, LRNNs delivered state-of-the-art image reconstruction quality, outperforming SPDER and SIREN even when using their respective component activations. They also produced superior audio fidelity with faster convergence and yielded higher-quality, artifact-free CT reconstructions.

Exciting avenues for future research include extending LRNNs to domains such as video modeling and unsteady PDEs. We view 3D scene reconstruction (NeRFs) as a particularly promising direction; we hypothesize that the multiplicative structure of LRNNs is naturally suited for capturing the high-frequency, view-dependent effects. Additionally, experiments in Appendix I on classification benchmarks suggest that LRNNs' applicability extends beyond continuous signal representation.

While remarkably effective, LRNNs present opportunities for further refinement. For instance, while our use of forward-mode AD proved highly efficient for Laplacian computations in PDE tasks, the general backward pass currently incurs a higher memory footprint than standard MLPs due to intermediate product storage. However, as detailed in Appendix B.2, strategies such as kernel fusion and mixed-precision training offer clear paths to mitigate this. In summary, LRNNs provide a versatile and powerful building block for learning compact and expressive representations across a broad spectrum of machine learning challenges.

ACKNOWLEDGEMENTS

This work is supported by a Natural Sciences and Engineering Research Council of Canada (NSERC) Discovery Grant.

REPRODUCIBILITY STATEMENT

We provide complete architectural specifications, hyperparameters, and training procedures in Sections 3 and 4 and Appendices B-I. Our implementation is publicly available at `https://github.com/dacelab/lrnn`. All experimental configurations, including learning rates, and model architectures for each benchmark, are detailed in the respective appendices.

ETHICS STATEMENT

Our work introduces a new deep learning architecture with demonstrated advantages in medical imaging (CT reconstruction) and scientific computing (numerical solution of PDEs). While these applications have clear benefits—reducing patient radiation exposure and advancing computational science—we acknowledge that powerful representation learning tools can be misused. The improved efficiency of LRNNs could potentially lower barriers to applications requiring careful ethical consideration. We encourage responsible deployment with appropriate domain expertise and ethical oversight, particularly in medical applications.

During the preparation of this work, the authors used LLMs for proofreading the text and for debugging code.

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

# A   THEORETICAL ANALYSIS OF LRNNS

## A.1   LRNN MODEL STRUCTURE

Consider an LRNN with one layer and separation-rank $r$, i.e., $f(\mathbf{x}) = \sum_{\ell=1}^{r} s_\ell \prod_{j=1}^{d}(1 + \gamma\, g_j^\ell(x_j))$, where $\mathbf{x} \in [0,1]^d$ and each $g_j^\ell : [0,1] \to \mathbb{R}$ is a univariate function. The factors $(1 + \gamma\, g_j^\ell(x_j))$ in our LRNN model explicitly separate a constant offset of 1 from the learnable component $\gamma\, g_j^\ell(x_j)$. This is a specific parameterization of the more general multiplicative factors in classical separated-rank decompositions. While this choice does not change the fundamental expressive power, it yields tangible advantages in model initialization, learning dynamics, and interpretation, as we detail next. For simplicity, we will drop the $\gamma$ scaling in the following discussion – we will study the role of $\gamma$ in Section A.3.

1. Built-in additive terms. Expanding $\prod_{j=1}^{d}(1 + g_j^\ell(x_j))$ automatically includes (i) a constant offset 1, (ii) purely additive terms $\sum_j g_j^l(x_j)$, (iii) all possible higher-order products. Although one can mimic this in a purely multiplicative model by adding a constant or bias term inside each factor, the $(1 + g_j)$ notation makes these offsets more explicit.

2. Initialization and automatic relevance determination. Setting each $g_j^\ell(\cdot)$ to zero at initialization yields an initial product of 1, a natural baseline. The network can then gradually learn interactions by adapting specific $g_j^\ell(\cdot)$ functions. If a feature $x_j$ is unimportant, the network can keep $g_j^\ell(\cdot) \approx 0$, which in our experience often leads to more stable training while providing an in-built automatic relevance determination mechanism.

3. Connection to functional ANOVA decompositions. The expanded form

$$\prod_{j=1}^{d}(1 + g_j^\ell(x_j)) = 1 + \sum_{j=1}^{d} g_j^\ell(x_j) + \sum_{j<k} g_j^\ell(x_j)g_k^\ell(x_k) + \cdots + \prod_{j=1}^{d} g_j^\ell(x_j)$$

is reminiscent of a *functional ANOVA* decomposition. In principle, a single rank-1 factor can capture constant, additive, pairwise, and all higher-order interactions. Rank-r LRNN is equivalent to a functional ANOVA decomposition whose component functions are weighted linear combinations of products of univariate functions.

We discuss below some additional reasons why the $(1 + \cdot)$ factorization can be advantageous in practice, even though it is not a fundamentally different decomposition than classic CP.

**Modeling sparse interactions.**   Suppose we want $f$ to capture a product of only a few relevant coordinates (e.g., $\prod_{j \in S} x_j$ for some small subset $S$). In a *purely* multiplicative model $\prod_{j=1}^{d} h_j(x_j)$, one typically sets $h_j(x_j) = x_j$ for $j \in S$ and $h_j(x_j) = 1$ for $j \notin S$. With LRNNs, we can do the same by setting $g_j^l(x_j) = x_j - 1$ for $j \in S$ and $g_j^\ell(x_j) = 0$ for $j \notin S$. Hence, the factor $\prod_{j=1}^{d}[1 + g_j^\ell(x_j)]$ becomes $\prod_{j \in S} x_j$, while all factors corresponding to irrelevant coordinates default to 1. This offset-based parameterization can be more natural to train or initialize.

**Compact representation of polynomial features.**   One can also view $(1 + g_j^\ell(x_j))$ as a generating function in each coordinate. When $g_j^\ell$ is a complete polynomial, the product term when expanded yields constant, linear, and higher-order powers of $x_j$. A classical fact (see, e.g., Kolda & Bader (2009) for a tensor viewpoint) is that polynomials with *fully factorable* coefficient structure can be captured in a single rank-1 product. Although this observation is not novel (it dates back to standard generating-function ideas and CP decompositions), it illustrates how $(1 + g_j^\ell)$ can unify constant, additive, and multiplicative terms in one factor.

In summary, the $(1 + \gamma\, g_j^\ell(\cdot))$ construction is mostly for notational and practical convenience, rather than a departure from classical multiplicative low-rank expansions. From a theoretical perspective, an LRNN is equivalent in representational power to a standard rank-$r$ CP model $\sum_{\ell=1}^{r} \prod_{j=1}^{d} h_j^\ell(x_j)$ that permits each $h_j^l$ to have a constant offset. However, specifying these offsets explicitly by $(1 + g_j^\ell)$ often simplifies initialization (starting from a constant baseline), captures additive terms by default, and can improve interpretability regarding how interactions are learned during training. Later in this

section, we will provide some theoretical analysis to show how the normalization constant $\gamma$ ensures that the variance of the LRNN output is controlled, which can be beneficial for training.

The deep LRNN architecture is illustrated in Figure 10.

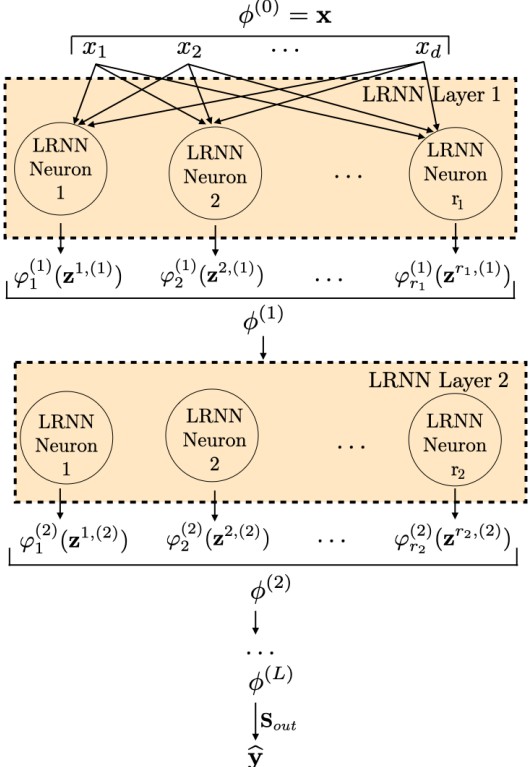

Figure 10: Deep LRNN architecture.

## A.2 INTERPRETABILITY AND INTERACTION ANALYSIS

A significant challenge in deep learning is the opaque nature of standard architectures. In a standard MLP, feature interactions are entangled within dense matrix multiplications, making it difficult to isolate the contribution of specific variables. The LRNN architecture, by virtue of its separable product structure, offers a more structured view of how features are combined, similar in spirit to generalized additive models (GAMs) such as neural additive models (NAMs) proposed by Agarwal et al. (2021).

GAMs approximate a function as a sum of univariate functions: $f(\mathbf{x}) = \sum f_j(x_j)$. LRNNs generalize this by allowing for multiplicative interactions. Each LRNN neuron $\ell$ computes a product of univariate transformations on projected features: $\phi_\ell(\mathbf{z}) = \prod_{j=1}^{\bar{d}}(1 + \gamma g_j^\ell(z_j))$. Since the component functions $g_j^\ell : \mathbb{R} \to \mathbb{R}$ are univariate, they can be visualized directly. By plotting the learned curves $g_j^\ell(z)$, practitioners can inspect the nonlinear transformation applied to each projected feature dimension (e.g., whether the model has learned an approximately linear trend, a threshold-like response, or a periodic modulation) before these features are combined multiplicatively.

Beyond visualizing individual components, the LRNN structure enables an explicit, architecture-level notion of which features participate in a given interaction. In the product above, each factor $\left(1 + \gamma g_j^\ell(z_j)\right)$ acts as a modulator. If, over the data distribution, a component function learns a nearly constant mapping $g_j^\ell(z) \approx 0$, then the corresponding factor is close to 1 and effectively does not influence the product. Conversely, if $g_j^\ell(z)$ varies substantially, it actively modulates the neuron output.

This observation suggests a simple diagnostic: by computing the empirical variance of each component function, $\mathrm{Var}[g_j^\ell(z_j)]$, over a validation set, one can construct "interaction heatmaps" that indicate which coordinates are active for each neuron. For example, if a specific neuron $\ell$ exhibits high variance predominantly for components $j = 1$ and $j = 4$, this provides evidence that this neuron is primarily sensitive to joint variation in feature dimensions 1 and 4 and relatively invariant to the others.

Standard MLPs with conventional activation functions can, in principle, be analyzed using post-hoc attribution or sensitivity methods, but their parameterization does not make such interaction structure explicit. In contrast, LRNNs build this structure into the architecture: the univariate components $g_j^\ell$ and their variances provide a direct handle for probing which dimensions and interactions a given neuron is using.

### A.3    VARIANCE–CONTROLLED INITIALIZATION: PROOF OF LEMMA 1

In this section, we provide a proof for Lemma 1 in Section 3.1 and discuss its implications. The proof is broken down into a lemma (see Lemma 3) and a corollary (see Corollary 1). An LRNN neuron acting on the $\bar{d}$-dimensional projection $\mathbf{z} = (z_1, \ldots, z_{\bar{d}})$ is defined by the product–structured activation $\varphi(\mathbf{z}) = \prod_{j=1}^{\bar{d}}(1 + \gamma\, g_j(z_j))$, where $\gamma = \frac{1}{\sqrt{\bar{d}}}$. Let $z_j = \mathbf{w}_j^\top \mathbf{x} + b_j$, with weights initialized such that for a fixed input $\mathbf{x}$, $\mathbb{E}[z_j] = 0$ and $\mathrm{Var}[z_j] = \sigma_z^2$ for each $j$. In addition, let each $g_j : \mathbb{R} \to \mathbb{R}$ be a twice-differentiable, learnable univariate function. We theoretically analyze the role of the scaling parameter $\gamma$ under the following standard assumptions at initialization:

**Assumption 1** (Independence across dimensions). *For any distinct indices $j, k \in \{1, \ldots, \bar{d}\}$, the random variables $z_j$ and $z_k$ are independent. In other words, $g_j(z_j)$ is independent of $g_k(z_k)$, and $g_j'(z_j)$ is independent of $g_k'(z_k)$ for $j \neq k$, as $g_j$ are deterministic functions of $z_j$ at initialization.*

**Assumption 2** (Properties of component functions). *For every $j \in \{1, \ldots, \bar{d}\}$, $\mathbb{E}[g_j(z_j)] = 0$, $\mathrm{Var}[g_j(z_j)] = \sigma_g^2$, $\mathbb{E}[g_j'(z_j)] = 0$, and $\mathrm{Var}[g_j'(z_j)] = \sigma_{g'}^2$, where $\sigma_g$ and $\sigma_{g'}$ are finite constants.*

Assumption 2 ensures that the expected partial derivatives $\mathbb{E}[\partial\varphi/\partial z_k]$ are zero, simplifying variance calculations. We note that while the main conclusion regarding the sum of gradient variances being $\mathcal{O}(1)$ holds more broadly (see discussion after Lemma 3), these specific assumptions lead to the following variance bounds.

**Lemma 3** (Gradient-variance stabilization). *Under Assumptions 1 and 2, the partial derivatives of the LRNN product-structured activation $\varphi(\mathbf{z}) = \prod_{j=1}^{\bar{d}}(1 + \gamma\, g_j(z_j))$, where $\gamma = \frac{1}{\sqrt{\bar{d}}}$ satisfy*

$$\mathbb{E}\left[\frac{\partial\varphi}{\partial z_k}\right] = 0, \quad and \quad \mathbb{E}\left[\left(\frac{\partial\varphi}{\partial z_k}\right)^2\right] \leq \frac{\sigma_{g'}^2}{\bar{d}}\, e^{\sigma_g^2} \quad \forall k \in \{1, \ldots, \bar{d}\}.$$

*and consequently the sum of the variances of the first-order partial derivatives satisfies the following bound that is independent of $\bar{d}$, i.e., $\sum_{k=1}^{\bar{d}} \mathrm{Var}\left[\frac{\partial\varphi}{\partial z_k}\right] \leq \sigma_{g'}^2\, e^{\sigma_g^2}$.*

*Proof.* From the definition of $\varphi$, we have $\partial\varphi/\partial z_k = \gamma\, g_k'(z_k) \prod_{j\neq k}(1 + \gamma\, g_j(z_j))$. Using Assumption 1 to separate expectations and Assumption 2 for the properties of $g_j$ and $g_j'$, we have

$$\mathbb{E}\left[\frac{\partial\varphi}{\partial z_k}\right] = \gamma\, \mathbb{E}[g_k'(z_k)] \prod_{j\neq k} \mathbb{E}[1 + \gamma\, g_j(z_j)] = \gamma \cdot 0 \cdot \prod_{j\neq k}(1 + \gamma \cdot 0) = 0.$$

For the second moment, we have

$$\mathbb{E}\left[\left(\frac{\partial\varphi}{\partial z_k}\right)^2\right] = \gamma^2\, \mathbb{E}[g_k'(z_k)^2] \prod_{j\neq k} \mathbb{E}[(1 + \gamma\, g_j(z_j))^2]$$

$$= \gamma^2\, (\mathrm{Var}[g_k'(z_k)] + (\mathbb{E}[g_k'(z_k)])^2) \prod_{j\neq k}(1 + 2\gamma\mathbb{E}[g_j(z_j)] + \gamma^2\mathbb{E}[g_j(z_j)^2])$$

$$= \frac{1}{\bar{d}}(\sigma_{g'}^2 + 0^2) \prod_{j\neq k}(1 + 0 + \frac{1}{\bar{d}}(\mathrm{Var}[g_j(z_j)] + (\mathbb{E}[g_j(z_j)])^2)) = \frac{1}{\bar{d}}\sigma_{g'}^2 \left(1 + \frac{\sigma_g^2}{\bar{d}}\right)^{\bar{d}-1}.$$

Using the inequality $(1 + \frac{a}{m})^{m-1} \leq e^a$ for $a, m > 0$[3], we have $\mathbb{E}\left[\left(\frac{\partial \varphi}{\partial z_k}\right)^2\right] \leq (\sigma_{g'}^2/\bar{d}) \, e^{\sigma_g^2}$. Since $\mathbb{E}[\partial \varphi/\partial z_k] = 0$, it follows that $\mathrm{Var}[\partial \varphi/\partial z_k] = \mathbb{E}[(\partial \varphi/\partial z_k)^2]$. Summing this variance bound over $k = 1, \ldots, \bar{d}$ yields the following upper bound independent of the projection width $\bar{d}$.

$$\sum_{k=1}^{\bar{d}} \mathrm{Var}[\partial \varphi/\partial z_k] \leq \sum_{k=1}^{\bar{d}} \frac{\sigma_{g'}^2}{\bar{d}} \, e^{\sigma_g^2} = \sigma_{g'}^2 \, e^{\sigma_g^2}$$

$\square$

**Remark 1** (Relaxing Assumption 2). *If $\mathbb{E}[g_j'(z_j)] = \mu_{g'} \neq 0$, then $\mathbb{E}[\partial \varphi/\partial z_k] = \gamma \mu_{g'}$. The sum of variances $\sum_k \mathrm{Var}[\partial \varphi/\partial z_k]$ would then be bounded by $[(\sigma_{g'}^2 + \mu_{g'}^2)e^{\sigma_g^2} - \mu_{g'}^2]$, which is still an $\mathcal{O}(1)$ constant independent of $\bar{d}$. Thus, the primary conclusion of width-independent total gradient variance still holds, though Assumption 2 simplifies the constant and ensures zero-mean gradients.*

We now prove a corollary showing that the variance of the output of the LRNN neuron remains $\mathcal{O}(1)$ as $\bar{d}$ increases.

**Corollary 1** (Activation variance). *Under Assumptions 1 and 2, the variance of the LRNN activation satisfies the inequality $\mathrm{Var}[\varphi(\mathbf{z})] \leq e^{\sigma_g^2} - 1$.*

*Proof.* Using Assumptions 1 and 2, we have $\mathbb{E}[\varphi(\mathbf{z})] = \prod_{j=1}^{\bar{d}} \mathbb{E}[1 + \gamma \, g_j(z_j)] = \prod_{j=1}^{\bar{d}}(1 + \gamma \cdot 0) = 1$. For the second moment, it follows from Assumption 2 that $\mathbb{E}[g_j(z_j)^2] = \sigma_g^2$, which gives

$$\mathbb{E}[\varphi(\mathbf{z})^2] = \prod_{j=1}^{\bar{d}} \mathbb{E}[(1 + \gamma \, g_j(z_j))^2] = \prod_{j=1}^{\bar{d}}(1 + 2\gamma\mathbb{E}[g_j(z_j)] + \gamma^2\mathbb{E}[g_j(z_j)^2])$$

$$= \prod_{j=1}^{\bar{d}}(1 + \gamma^2\sigma_g^2) = \left(1 + \frac{\sigma_g^2}{\bar{d}}\right)^{\bar{d}}.$$

Therefore, $\mathrm{Var}[\varphi(\mathbf{z})] = \mathbb{E}[\varphi(\mathbf{z})^2] - (\mathbb{E}[\varphi(\mathbf{z})])^2 = \left(1 + \frac{\sigma_g^2}{\bar{d}}\right)^{\bar{d}} - 1$. Using the inequality $(1 + a/m)^m \leq e^a$ that holds for $a, m > 0$, yields the stated result. $\square$

**Remark 2** (Implications). *Lemma 3 demonstrates that while the variance of each individual coordinate-gradient $\partial \varphi/\partial z_k$ decays like $1/\bar{d}$, their cumulative variance sum remains constant. This suggests an intrinsic mechanism for automatic relevance determination: as projection width $\bar{d}$ grows, the influence of any single projected coordinate $z_k$ on the output's gradient variance diminishes. Together with Corollary 1, which ensures $\mathcal{O}(1)$ activation variance, this analysis establishes that the scaling factor $\gamma = 1/\sqrt{\bar{d}}$ plays a crucial role analogous to initialization approaches for additive NNs (Glorot & Bengio, 2010; He et al., 2015) or the $1/\sqrt{r}$ scaling in LoRA adapters (Hu et al., 2022), ensuring stable propagation in both forward and backward passes for LRNN neurons, regardless of product width.*

### A.4 UNIVERSAL APPROXIMATION THEOREM: PROOF OF THEOREM 1

**Theorem 1** (Universal approximation theorem). *If $f : [0, 1]^d \to \mathbb{R}$ is a continuous function, then for every $\varepsilon > 0$, there exists an LRNN model $\widehat{f}_{\mathrm{lrnn}}(\mathbf{x}) = \sum_{\ell=1}^{r} s_\ell \prod_{j=1}^{d}(1 + g_j^\ell(x_j))$ with suitably chosen separation rank $r \leq R(\varepsilon)$ and univariate component functions $g_j^l : [0, 1] \to \mathbb{R}$ such that*

$$\max_{\mathbf{x} \in [0,1]^d} \left| f(\mathbf{x}) - \widehat{f}_{\mathrm{lrnn}}(\mathbf{x}) \right| \leq \varepsilon.$$

*Proof.* We provide a proof based on classical tensor-product expansions and polynomial approximations; see, for example, Hornik (1991); Pinkus (1999); Cybenko (1989) for analogous MLP proofs.

---

[3]This follows from the fact that $(1 + \frac{a}{m})^{m-1} = \frac{(1+a/m)^m}{1+a/m} \leq \frac{e^a}{1}$ as $1 + a/m \geq 1$.

It follows from the Stone-Weierstrass theorem that any continuous function on the compact domain $[0,1]^d$ can be uniformly approximated by a multivariate polynomial. Thus, there exist $N \in \mathbb{N}$, real coefficients $a_\alpha$, and univariate polynomials $\{\phi_{j,\alpha}(x_j) : [0,1] \to \mathbb{R}\}_{j=1}^d$, such that $\widehat{f}(\mathbf{x}) = \sum_{\ell=1}^N a_\ell \prod_{j=1}^d \phi_{j,\ell}(x_j)$, with $\sup_{\mathbf{x} \in [0,1]^d} |f(\mathbf{x}) - \widehat{f}(\mathbf{x})| \leq \varepsilon/2$

Noting that $\sum_{\ell=1}^N a_\ell \prod_{j=1}^d \phi_{j,\ell}(x_j)$ is a sum of product terms representation, we will set $r := N$, $s_\ell := a_\ell$, and define the univariate functions $g_j^\ell(x_j) := \phi_{j,\ell}(x_j) - 1$ in the LRNN representation such that

$$\sum_{\ell=1}^r s_\ell \prod_{j=1}^d \left[1 + g_j^\ell(x_j)\right] = \sum_{\ell=1}^r a_\ell \prod_{j=1}^d \phi_{j,\ell}(x_j).$$

It follows from standard universal approximation results (see, for example Hornik (1991)) that each component function $g_j^\ell$ can be approximated by a univariate neural network $\widetilde{g}_j^\ell(x_j)$ such that $\max_{x_j \in [0,1]} |g_j^\ell(x_j) - \widetilde{g}_j^\ell(x_j)| \leq \delta$. Using a telescoping product approach yields the inequality

$$\left| \sum_{\ell=1}^r s_\ell \prod_{j=1}^d (1 + g_j^\ell(x_j)) - \sum_{\ell=1}^r s_\ell \prod_{j=1}^d (1 + \widetilde{g}_j^\ell(x_j)) \right| \leq rd S_{\max} M^{d-1} \delta,$$

where $M := \max_{j,\ell} \sup_{z \in [0,1]} |1 + g_j^\ell(z)|$ is bounded by construction and $S_{\max} = \max |s_\ell|$. Choosing $\delta \leq \varepsilon/(2rd S_{\max} M^{d-1})$ completes the proof. $\qquad \square$

## A.5 STRUCTURED FUNCTIONS WITH DECAYING ANOVA INTERACTIONS: PROOF OF THEOREM 2

We show that LRNNs can mitigate the curse of dimensionality for a class of high-dimensional functions whose functional ANOVA decomposition exhibits a decay in the importance of higher-order interaction terms, and whose significant low-order terms are themselves approximable by sums of factorized components. The analysis presented here focuses on the case when the LRNN component functions are univariate MLPs. To prove our main result, we first establish a lemma on the approximation of functions that are already sums of products of univariate functions, where each product involves a limited number of variables.

**Lemma 4** (LRNN approximation of separable low-rank functions). *Let $f : [0,1]^d \to \mathbb{R}$ denote a continuous function that admits the representation $f(\mathbf{x}) = \sum_{\ell=1}^r \alpha_\ell \prod_{j \in S_\ell} \Phi_{j,\ell}(x_j)$, where $S_\ell \subseteq \{1, 2, \ldots, d\}$ with $|S_\ell| \leq m$ for some $m \leq d$, $\Phi_{j,\ell} : [0,1] \to \mathbb{R}$, $j \in S_\ell$ are continuous univariate functions such that $|\alpha_\ell| \leq c$ and $\|\Phi_{j,\ell}\|_\infty \leq 1$, where $c \geq 0$ is a constant. Then for any $\varepsilon > 0$, there exists an LRNN approximation, $\widehat{f}_{\mathrm{lrnn}}$, with $\mathcal{O}(rm^2/P(c,r,m,\varepsilon))$ parameters such that $\sup_{\mathbf{x} \in [0,1]^d} |f(\mathbf{x}) - \widehat{f}_{\mathrm{lrnn}}(\mathbf{x})| \leq \varepsilon$, where $P(c,r,m,\varepsilon) = \ln(1 + \frac{\varepsilon}{rc})$.*

*Proof.* We begin by noting that the target function admits exact representation by an LRNN by appropriately defining the scale and univariate component functions, i.e., $f(\mathbf{x}) = f_{\mathrm{lrnn}}(\mathbf{x}) = \sum_{\ell=1}^r s_\ell \prod_{j=1}^d (1 + \widetilde{g}_j^\ell(x_j))$ with $s_\ell = \alpha_\ell$ and $\widetilde{g}_j^\ell(x_j) = \Phi_{j,\ell}(x_j) - 1 \; \forall j \in S_\ell$ and $\widetilde{g}_j^\ell = 0 \; \forall j \notin S_\ell$.

Each non-trivial $\widetilde{g}_j^\ell$ is approximated by a univariate MLP $g_j^\ell : \mathbb{R} \to \mathbb{R}$ with sufficient capacity such that $\|g_j^\ell - \widetilde{g}_j^\ell\|_\infty \leq \delta$. It follows from standard universal approximation results (Hornik, 1991; Pinkus, 1999) that the number of parameters needed for each $g_j^\ell$ is $\mathcal{O}(1/\delta)$.

Since $\|\Phi_{j,\ell}\|_\infty \leq 1$, we have $|1 + \widetilde{g}_j^\ell(x_j)| \leq 1$, and $|1 + g_j^\ell(x_j)| = |1 + \widetilde{g}_j^\ell(x_j) + g_j^\ell(x_j) - \widetilde{g}_j^\ell(x_j)| \leq 1 + \delta \; \forall j \in S_\ell, x_j \in [0,1]$, while $|1 + \widetilde{g}_j^\ell(x_j)| = 1$ and $|1 + g_j^\ell(x_j)| = 1 \; \forall j \notin S_\ell, x_j \in [0,1]$ (since $\widetilde{g}_j^\ell = 0$ for $j \notin S_\ell$).

Let $k_\ell = |S_\ell| \leq m$ be the number of interacting variables in the $\ell$-th term. To bound the error between $f$ and $\widehat{f}_{\mathrm{lrnn}}$, we first consider the error for the $\ell$-th term in the sum, i.e., $E_\ell := |\prod_{j \in S_\ell}(1 + g_j^\ell(x_j)) - \prod_{j \in S_\ell}(1 + \widetilde{g}_j^\ell(x_j))|$. Let $\{p_1, p_2, \ldots, p_{k_\ell}\}$ be an ordered enumeration of the indices in

$S_\ell$. Using a telescoping product argument, we have

$$E_\ell \leq \sum_{k=1}^{k_\ell} \left| g_{p_k}^\ell(x_{p_k}) - \widetilde{g}_{p_k}^\ell(x_{p_k}) \right| \prod_{i=1}^{k-1} \left| 1 + g_{p_i}^\ell(x_{p_i}) \right| \prod_{i=k+1}^{k_\ell} \left| 1 + \widetilde{g}_{p_i}^\ell(x_{p_i}) \right|$$

$$\leq \sum_{k=1}^{k_\ell} \delta \cdot (1+\delta)^{k-1} = \delta \sum_{j=0}^{k_\ell-1} (1+\delta)^j = (1+\delta)^{k_\ell} - 1.$$

Since $k_\ell \leq m$, we have $E_\ell \leq (1+\delta)^m - 1$. The total approximation error can be bounded as:

$$\sup_{\mathbf{x} \in [0,1]^d} \left| f(\mathbf{x}) - \widehat{f}_{\text{lrnn}}(\mathbf{x}) \right| \leq \sum_{\ell=1}^r |\alpha_\ell| \, E_\ell \leq \sum_{\ell=1}^r c\left((1+\delta)^m - 1\right) = rc\left((1+\delta)^m - 1\right).$$

We need to choose $\delta$ such that the total approximation error to be less than or equal to $\varepsilon$, i.e., $rc((1+\delta)^m - 1) \leq \varepsilon$. Using the inequality $(1+y)^k \leq e^{yk}$ for $y \geq 0, k \geq 1$ with $y = \delta$ and $k = m$ in the preceding equation, we have $rc\left(e^{m\delta} - 1\right) \leq \varepsilon$. Taking the logarithm of both sides and rearranging gives $\delta = (1/m)\ln\left(1 + \frac{\varepsilon}{rc}\right)$. Noting that we have at most $rm$ non-trivial univariate functions, with each requiring $\mathcal{O}(1/\delta)$ parameters, we obtain the stated parameter complexity. $\square$

**Remark 3** (Simplified parameter complexity). *If $\varepsilon \ll rc$ the parameter count simplifies to $\mathcal{O}(r^2 m^2 c/\varepsilon)$. The parameter complexity grows as $m^2$ ($m \leq d$) for fixed $(r, c)$, thereby circumventing the curse of dimensionality.*

**Remark 4** (Assumption $\|\Phi_{j,\ell}\|_\infty \leq 1$). *The assumption $\|\Phi_{j,\ell}\|_\infty \leq 1$ is a common approach to normalize components in approximation theory. In practical machine learning scenarios, this is often justified since the target function $f(\mathbf{x})$ is typically normalized. The constants $r$ and $c$ (governing $|\alpha_\ell|$) characterize a specific sum-of-products decomposition assumed to exist for this normalized target function, where its constituent univariate functions $\Phi_{j,\ell}$ have norms bounded by 1. If such a representation exists for the normalized target, this condition is met.*

We now prove our main theorem for a general continuous function $f : [0,1]^d \to \mathbb{R}$ whose functional ANOVA decomposition takes the form:

$$f(\mathbf{x}) = \sum_{S \subseteq [d]} f_S(\mathbf{x}_S), \tag{6}$$

where each $f_S$ depends only on variables $\mathbf{x}_S$ with $S \subseteq \{1, \ldots, d\}$ and $\mathbf{x}_S := (x_j)_{j \in S}$. We assume that standard ANOVA orthogonality conditions hold, e.g., $\int_0^1 f_S(\mathbf{x}_S) dx_j = 0$ for any $j \in S$ and $f_\emptyset = \int f(\mathbf{x}) d\mathbf{x}$. Our theoretical analysis uses the following assumptions:

**Assumption 3** (Decay of ANOVA components). *The norms of ANOVA components decay sufficiently fast. Specifically, for a given $\varepsilon_1 > 0$, let $\mathcal{I}_{trunc} = \{S \subseteq [d] : \|f_S\|_\infty \geq \tau_S\}$ be a collection of index sets such that the truncated sum $f_{\text{trunc}}(\mathbf{x}) = \sum_{S \in \mathcal{I}_{trunc}} f_S(\mathbf{x}_S)$ satisfies $\|f - f_{\text{trunc}}\|_\infty \leq \varepsilon_1$. Let $N_a = |\mathcal{I}_{trunc}|$ be the number of significant ANOVA terms, and let $m_a = \max_{S \in \mathcal{I}_{trunc}} |S|$ be their maximum interaction order. In addition, let $m_a \ll d$, and let $N_a$ grow at most polynomially with $d$.*

**Assumption 4** (Factorizability of ANOVA components). *Each ANOVA component can be approximated by a sum-product representation. Specifically, for each ANOVA component $f_S(\mathbf{x}_S)$ $\forall S \in \mathcal{I}_{trunc}$, there exists $\varepsilon_2 > 0$ such that the sum-product representation $\widehat{f}_S(\mathbf{x}_S) = \sum_{\beta=1}^{k_S} \alpha_{S,\beta} \prod_{j \in S} \phi_{j,S,\beta}(x_j)$ satisfies the error bound $\|f_S - \widehat{f}_S\|_\infty \leq \varepsilon_2$, $k_S \leq k_{max}$ for some $k_{max} \geq 1$, $|\alpha_{S,\beta}| \leq c$ and $\|\phi_{j,S,\beta}\|_\infty \leq 1$.*

**Remark 5** (The assumption $\|\phi_{j,S,\beta}\|_\infty \leq 1$). *Each significant ANOVA term $f_S(\mathbf{x}_S)$ is itself a function of at most $m_a$ variables. If the target function is pre-normalized, its ANOVA components $f_S$ (which are defined through integrals of $f$) also inherit scaling properties. It is then plausible that these (scaled) $f_S$ terms can be well-approximated by a sum-of-products representation where the individual univariate components $\phi_{j,S,\beta}$ are also normalized. The practical strength of this assumption rests on whether such a 'normalized-component' sum-of-products approximation for each relevant $f_S$ exists with a simultaneously controlled number of terms and bounded coefficients. For ANOVA terms $f_S$ that are smooth and depend on a small number of variables (small $m_a$), this is often considered a reasonable modeling assumption.*

**Theorem 2.** *If $f : [0, 1]^d \to \mathbb{R}$ satisfies Assumption 3 and Assumption 4, then for any target accuracy $\varepsilon > 0$, there exists an LRNN approximation, $\widehat{f}_{\mathrm{lrnn}}$, with parameter complexity $\mathcal{O}(\frac{(N_a k_{\max})^2 m_a^2 c}{\varepsilon})$ such that $\sup_{\mathbf{x} \in [0,1]^d} |f(\mathbf{x}) - \widehat{f}_{\mathrm{lrnn}}(\mathbf{x})| \le \varepsilon$.*

*Proof.* It follows from Assumption 3 that truncating the ANOVA expansion of $f$ gives $f(\mathbf{x}) \approx f_{\mathrm{trunc}}(\mathbf{x}) = \sum_{S \in \mathcal{I}_{\mathrm{trunc}}} f_S(\mathbf{x}_S)$ such that $\|f - f_{\mathrm{trunc}}\|_\infty \le \varepsilon_1$. From Assumption 4, we can approximate $f_S$ as $\widehat{f}_S(\mathbf{x}_S) = \sum_{\beta=1}^{k_S} \alpha_{S,\beta} \prod_{j \in S} \phi_{j,S,\beta}(x_j)$ such that the error is $\|f_S - \widehat{f}_S\|_\infty \le \varepsilon_2$.

Let $h(\mathbf{x}) = \sum_{S \in \mathcal{I}_{\mathrm{trunc}}} \widehat{f}_S(\mathbf{x}_S)$. Then $\|f_{\mathrm{trunc}} - h\|_\infty \le \sum_{S \in \mathcal{I}_{\mathrm{trunc}}} \|f_S - \widehat{f}_S\|_\infty \le N_a \varepsilon_2$. We now approximate $h$ using an LRNN, $\widehat{f}_{\mathrm{lrnn}}$, such that $\|h - \widehat{f}_{\mathrm{lrnn}}\|_\infty \le \varepsilon_{lrnn}$. The total error is $\|f - \widehat{f}_{\mathrm{lrnn}}\|_\infty \le \varepsilon_1 + N_a \varepsilon_2 + \varepsilon_{\mathrm{lrnn}}$. We set $\varepsilon_1 = \varepsilon/3$, $N_a \varepsilon_2 = \varepsilon/3$ (i.e., $\varepsilon_2 = \varepsilon/(3N_a)$), and $\varepsilon_{\mathrm{lrnn}} = \varepsilon/3$ to ensure that the total error is at most $\varepsilon$.

To bound the parameter complexity of $\widehat{f}_{\mathrm{lrnn}}$, we note that

$$h(\mathbf{x}) = \sum_{S \in \mathcal{I}_{trunc}} \sum_{\beta=1}^{k_S} \alpha_{S,\beta} \prod_{j \in S} \phi_{j,S,\beta}(x_j)$$

is a sum of products involving $\sum_{S \in \mathcal{I}_{trunc}} k_S \le N_a k_{\max}$ terms. Each product $\prod_{j \in S} \phi_{j,S,\beta}(x_j)$ involves $|S| \le m_a$ univariate functions with coefficients $\alpha_{S,\beta}$, where $|\alpha_{S,\beta}| \le c$. Moreover, the univariate functions $\phi_{j,S,\beta}$ satisfy the bound $\|\phi_{j,S,\beta}\|_\infty \le 1$. Thus, $h$ matches the form required by Lemma 4, with parameters: $r = R \le N_a k_{\max}$ and $m = m_a$. It therefore follows that $h$ can be approximated by an LRNN $\widehat{f}_{\mathrm{lrnn}}$ to an accuracy $\varepsilon_{\mathrm{lrnn}} = \varepsilon/3$, with the number of parameters (using the simplified parameter count) given by $\mathcal{O}(\frac{(N_a k_{\max})^2 m_a^2 c}{\varepsilon/3})$. Absorbing the constant $1/3$ into the $\mathcal{O}$ notation yields the stated complexity. $\square$

**Remark 6** (Special cases). *If $m_a$ and $k_{\max}$ are small constants (or grow very slowly with $d$), and $c$ is small, the complexity depends primarily on $N_a^2/\varepsilon$. If the ANOVA decay is such that $N_a \approx \sum_{k=0}^{m_a} \binom{d}{k} \approx \mathcal{O}(d^{m_a})$, then the parameter complexity becomes $\mathcal{O}((d^{2m_a} k_{\max}^2 m_a^2 c)/\varepsilon)$. This complexity is polynomial in $d$ if $m_a$ is constant, demonstrating mitigation of the exponential growth in parameter complexity.*

## A.6 REMARKS ON THEOREM 2

Theorem 2 establishes that if a high-dimensional function admits a functional ANOVA decomposition dominated by low-complexity terms, LRNNs can approximate it with a number of parameters that scales polynomially with the effective dimension $d$. To provide further insight, we examine the intuition behind Assumptions 3 and 4 and the specific function classes they encompass.

Assumption 3 requires that the number of significant ANOVA terms, $N_a$, grows at most polynomially with dimension $d$. This implicitly constrains the *maximum interaction order $m_a$*. Since a full ANOVA decomposition contains $2^d$ terms, for the truncated sum to scale polynomially (e.g., $N_a \approx \mathcal{O}(d^k)$), the interaction order $m_a$ must generally be small relative to $d$ (i.e., $m_a = \mathcal{O}(1)$). This characterizes functions with a low *effective superposition dimension* Caflisch et al. (1997), a property common in physical systems dominated by main effects and low-order (pairwise or triplet) interactions.

Assumption 4 posits that significant high-order interaction terms can be approximated by a tensor product with limited rank $k_{max}$. This can be viewed as a conservation of complexity principle; the curse of dimensionality is constrained rather than eliminated. Specifically, if ANOVA terms involve few variables ($m_a \ll d$), Assumption 4 is easily satisfied as low-dimensional functions typically admit efficient low-rank approximations. Conversely, if a term involves all variables ($m_a \approx d$), Assumption 4 requires it to be rank-deficient (small $k_{max}$).

Thus, LRNNs efficiently represent functions that are either *interaction-sparse* (small $m_a$) or *interaction-dense but rank-sparse* (small $k_{max}$). Theorem 2 implies that if a function has both global interactions ($m_a \approx d$) and high separation rank, $k_{max}$ would necessarily scale exponentially with $d$, reintroducing the curse of dimensionality.

- *Pairwise potentials (illustrates Assumption 3):* Functions such as Coulomb or gravitational potentials $V(\mathbf{x}) = \sum_{i \neq j} \phi(\|x_i - x_j\|)$ are dominated by pairwise interactions ($m_a = 2$). Even for large $d$, the number of significant ANOVA terms $N_a$ grows quadratically in $d$, satisfying the polynomial-growth requirement in Assumption 3.

- *Separable functions (illustrates Assumption 4):* A product state $f(\mathbf{x}) = \prod_{i=1}^{d} \sin(x_i)$ has maximum interaction order ($m_a = d$), so it does not have low interaction order. However, it is exactly rank-1 as a product of univariate factors ($k_{\max} = 1$), satisfying Assumption 4 with a single significant ANOVA term ($N_a = 1$). In this case the parameter bound in Theorem 2 still scales polynomially in $d$ because both $N_a$ and $k_{\max}$ are constants.

- *Sign-parity function (violates Assumptions 3 and 4):* The parity function $f(\mathbf{x}) = \prod_{i=1}^{d} \mathrm{sgn}(x_i)$ possesses both full interaction order ($m_a = d$) and high separation rank. It cannot be approximated by a small sum of smooth product terms and therefore falls outside the scope of Theorem 2.

In summary, when a function satisfies these structural assumptions (as in the first two examples), the LRNN architecture naturally aligns with its decomposition. As established in Lemma 4, this alignment allows LRNNs to learn the representation efficiently, achieving a parameter complexity that scales polynomially rather than exponentially with dimension. This theoretical result provides an insight into the empirical success of LRNNs on tasks where underlying low-order interaction structures are likely present.

### A.7 SPECTRAL REPRESENTATION ANALYSIS OF LRNNS: PROOF OF LEMMA 2

Here, we study the spectral representation capabilities of LRNNs, particularly when equipped with periodic activation functions commonly found in the INR literature, such as SIREN ($\sigma(z) = \sin(z)$) or SPDER (e.g., $\sigma(z) = \sin(z)\sqrt{|z|}$). We use $\omega$ to denote a scalar frequency parameter, which can be specific to each univariate function ($\omega_k$) or shared.

For simplicity, we do not consider the normalization term $\gamma$ in our analysis. We begin by recalling the observation in Section 3.1 that LRNNs with $\bar{d} = 1$ recover standard MLPs. Writing the LRNN activation function as $\varphi(\mathbf{x}) = \prod_{k=1}^{\bar{d}}(1 + g_k(z_k))$, where $z_k = \mathbf{v}_k^T \mathbf{x} + c_k$ and setting $\bar{d} = 1$, we have $\varphi(z_1) = 1 + g_1(z_1)$. Let $g_1(z_1) = \sigma(\omega_1 z_1)$ be a periodic activation function, where $z_1 = \mathbf{v}_1^T \mathbf{x} + c_1$. The full LRNN model is a sum of $r$ such rank-1 terms, each with its own scaling factor $s_\ell$: $f_{\mathrm{lrnn}}(\mathbf{x}) = \sum_{\ell=1}^{r} s_\ell \varphi_\ell(\mathbf{x}) = \sum_{\ell=1}^{r} s_\ell(1 + g_{1,\ell}(z_{1,\ell}))$. Substituting $g_{1,\ell}(z_{1,\ell}) = \sigma(\omega_{1,\ell}(\mathbf{v}_{1,\ell}^T \mathbf{x} + c_{1,\ell}))$ leads to

$$f_{\mathrm{lrnn}}(\mathbf{x}) = \sum_{\ell=1}^{r} s_\ell + \sum_{\ell=1}^{r} s_\ell\, \sigma(\omega_{1,\ell}(\mathbf{v}_{1,\ell}^T \mathbf{x} + c_{1,\ell})).$$

The second term in the preceding equation is an MLP with $r$ neurons in the hidden layer, where $\mathbf{v}_{1,\ell}$ are the input-to-hidden weights, $c_{1,\ell}$ are the hidden biases, $\sigma$ is the activation function (scaled by $\omega_{1,\ell}$), and $s_\ell$ are the hidden-to-output weights. The first term, $\sum_{\ell=1}^{r} s_\ell$, is a constant, acting as an overall output bias for the MLP. Thus, an MLP with a given periodic activation function can be viewed as a special case of an LRNN with $\bar{d} = 1$ and the same base activation. This observation forms the basis for comparing their spectral properties. The key distinction of LRNNs with $\bar{d} > 1$ lies in their product structure, which leads to a richer spectral synthesis than the purely additive nature of MLPs which we establish in the following lemma.

**Lemma 2** (Combinatorial frequency generation by LRNNs)**.** *Consider the LRNN product-structured activation function $\varphi(\mathbf{x}) = \prod_{k=1}^{\bar{d}}(1 + g_k(\mathbf{x}))$, where $g_k(\mathbf{x}) = \sigma_k(\omega_k(\mathbf{v}_k^T \mathbf{x} + c_k))$ is a univariate function with characteristic frequencies $\pm\mathbf{f}_k$ in the Fourier domain. Then, the Fourier transform of $\varphi(\mathbf{x})$ contains spectral components at not only the fundamental frequencies $\pm\mathbf{f}_k$ but also at all possible sum and difference combinations, i.e., frequencies of the form $\sum_{k \in S} s_k \mathbf{f}_k$ where $S \subseteq \{1, \ldots, \bar{d}\}$ and $s_k \in \{+1, -1\}$.*

*Proof.* The Fourier transform of $h_k(\mathbf{x}) = 1 + g_k(\mathbf{x})$ can be written as

$$\mathcal{F}\{h_k\}(\boldsymbol{\xi}) = \mathcal{F}\{1\}(\boldsymbol{\xi}) + \mathcal{F}\{g_k\}(\boldsymbol{\xi}) = \delta(\boldsymbol{\xi}) + G_k(\boldsymbol{\xi}), \tag{7}$$

where $\delta(\boldsymbol{\xi})$ is the Dirac delta representing the DC component (zero frequency), and $G_k(\boldsymbol{\xi})$ is the Fourier transform of $g_k(\mathbf{x})$, which by assumption has significant energy at $\pm\mathbf{f}_k$.

Since the LRNN activation is a product of these $h_k(\mathbf{x})$ terms, i.e., $\varphi(\mathbf{x}) = \prod_{k=1}^{\bar{d}} h_k(\mathbf{x})$, its Fourier transform takes the form

$$\mathcal{F}\{\varphi\}(\boldsymbol{\xi}) = \mathcal{F}\{h_1\} * \mathcal{F}\{h_2\} * \cdots * \mathcal{F}\{h_{\bar{d}}\} = \left( *_{k=1}^{\bar{d}} \mathcal{F}\{h_k\} \right)(\boldsymbol{\xi}),$$

where $*$ denotes convolution. Using equation 7 we have

$$\mathcal{F}\{\varphi\}(\boldsymbol{\xi}) = \left( \overset{\bar{d}}{\underset{k=1}{*}} (\delta(\cdot) + G_k(\cdot)) \right)(\boldsymbol{\xi}).$$

Now, expanding the product in the LRNN activation yields

$$\varphi(\mathbf{x}) = \prod_{k=1}^{\bar{d}} (1 + g_k(\mathbf{x})) = 1 + \sum_i g_i(\mathbf{x}) + \sum_{i<j} g_i(\mathbf{x})g_j(\mathbf{x}) + \cdots + \prod_{k=1}^{\bar{d}} g_k(\mathbf{x}).$$

The Fourier transform of the first term is $\mathcal{F}\{1\} = \delta(\boldsymbol{\xi})$ (the DC component). The second term, $\mathcal{F}\{\sum_i g_i(\mathbf{x})\} = \sum_i G_i(\boldsymbol{\xi})$, contributes the fundamental frequencies $\pm\mathbf{f}_i$ from each $g_i$. The third term, $\mathcal{F}\{\sum_{i<j} g_i(\mathbf{x})g_j(\mathbf{x})\} = \sum_{i<j}(G_i * G_j)(\boldsymbol{\xi})$ leads to pairwise combination frequencies, e.g., $\pm\mathbf{f}_i \pm \mathbf{f}_j$. This pattern continues for higher-order terms since a product of $p$ functions $g_{k_1}(\mathbf{x})\ldots g_{k_p}(\mathbf{x})$ will have a Fourier transform of the form $(G_{k_1} * \cdots * G_{k_p})(\boldsymbol{\xi})$, generating frequencies corresponding to all combinations $\pm\mathbf{f}_{k_1} \pm \cdots \pm \mathbf{f}_{k_p}$. Thus, a single LRNN activation with $\bar{d}$ components generates up to $2^{\bar{d}}-1$ distinct frequency combinations from just $\bar{d}$ base frequencies. $\quad\square$

Lemma 2 provides some useful insights into differences between how LRNNs and MLPs represent functions in the frequency domain. Standard MLPs (equivalent to LRNNs with $\bar{d} = 1$) synthesize functions *additively* in the spectral domain. Each neuron, with activation $s_l\sigma(\omega_l(\mathbf{v}_l^T\mathbf{x} + b_l))$, contributes primarily to a specific frequency pair $\pm\mathbf{f}_l = \pm\omega_l\mathbf{v}_l$ (for SIREN) or a narrow band around these (for SPDER-like activations). The overall spectrum of an MLP is the linear superposition of these individual contributions: $\mathcal{F}\{f_{MLP}\} = \sum_l s_l G_l(\boldsymbol{\xi})$. To represent a complex spectrum with many frequencies, an MLP typically requires a corresponding number of neurons. In contrast, LRNNs with $\bar{d} > 1$ employ a *multiplicative synthesis* within each rank-1 term. As shown in Lemma 2, a single rank-1 term can generate a combinatorial set of frequencies from just $\bar{d}$ base projected features $g_k$. The full LRNN then additively combines these rank-1 components, i.e., $\mathcal{F}\{f_{\text{lrnn}}\} = \sum_l s_l \mathcal{F}\{\varphi_l\}(\boldsymbol{\xi})$.

This combinatorial frequency synthesis mechanism provides LRNNs with adaptive spectral bias control—by learning the base frequencies $\mathbf{f}_k$ and their corresponding amplitudes, the network implicitly controls a rich set of derived frequencies, enabling efficient representation of complex spectra with fewer parameters than additive approaches.

## B  IMPLEMENTATION ASPECTS OF LRNNS

### B.1  PARAMETRIZATION OF LRNN COMPONENT FUNCTIONS

In our architecture, the univariate component functions $g_j^{\ell,(k)} : \mathbb{R} \to \mathbb{R}$ are parameterized by small, independent MLPs. Each such MLP, denoted $\text{MLP}(j, \ell, k)$, takes the scalar $z_j^{\ell,(k)}$ as its input, where $z_j^{\ell,(k)}$ is the $j$-th component of the vector $\mathbf{z}^{\ell,(k)} = \mathbf{W}^{\ell,(k)}\phi^{(k-1)} + \mathbf{b}^{\ell,(k)}$ resulting from the linear projection within the LRNN layer.

A key design choice for these component MLPs, $\text{MLP}(j, \ell, k)$, is that their first effective operation on the input $z_j^{\ell,(k)}$ is an activation function. Since $z_j^{\ell,(k)}$ is already the output of a linear transformation, applying another linear layer as the immediate first step within $\text{MLP}(j, \ell, k)$ would be redundant. Instead, the structure is:

$$g_j^{\ell,(k)}(z_j^{\ell,(k)}) = \text{MLP}(j, \ell, k)(z_j^{\ell,(k)}) \tag{8}$$

where the MLP $g_j^{\ell,(k)}$ is structured such that its computation begins with an initial activation $\sigma$, which is chosen to a standard activation or an INR style activation. The output of first stage of $\text{MLP}(j,\ell,k)$ is given by $\sigma(z_j^{\ell,(k)})$. This activated value is then processed by the subsequent hidden layers and the final output layer of $\text{MLP}(j,\ell,k)$ which is a linear layer without a bias term.

All learnable parameters, including those in $\mathbf{W}^{\ell,(k)}$, $\mathbf{b}^{\ell,(k)}$, and within all parts of $g_j^{\ell,(k)}$ are optimized jointly. The set of all univariate MLPs is implemented efficiently for parallel computation using block-diagonal weight matrices as a custom PyTorch layer. This design choice allows for efficient computation of the LRNN forward pass, as all the univariate functions in an LRNN layer can be computed in parallel.

## B.2 Memory Complexity Analysis and Optimization Strategies

While LRNNs demonstrate superior parameter efficiency and expressivity compared to standard architectures, the product-structured activation introduces a distinct memory footprint profile. In this section, we analyze the memory requirements for training LRNNs, focusing on the scaling behavior with respect to the projection width $\bar{d}$, and discuss strategies for optimized implementation.

To isolate the memory requirements when training LRNNs, consider a single LRNN layer with separation rank $r$ (analogous to $r$ neurons in a standard MLP) and projection width $\bar{d}$. We assume the univariate component functions $g_j^\ell(\cdot)$ are parameterized by a shared, shallow MLP with one hidden layer of width $h$, a setup consistent with our experimental configuration. Let $B$ denote the batch size. In a standard backpropagation framework (e.g., PyTorch Autograd), intermediate activations must be stored during the forward pass to compute gradients during the backward pass. We analyze the storage requirements for the activations (excluding model parameters, which are negligible in comparison).

For a standard MLP layer with $r$ neurons, the output takes the form $\mathbf{y} = \sigma(\mathbf{W}\mathbf{x} + \mathbf{b})$. The primary memory cost is storing the pre-activation or post-activation vectors, leading to a memory requirement that scales as $\mathcal{M}_{\text{MLP}} \approx \mathcal{O}(Br)$.

An LRNN neuron computes $\phi = \prod_{j=1}^{\bar{d}}(1 + \gamma g(z_j))$. The forward pass of this layer involves three distinct stages of activations that must be stored for backpropagation:

1. **Input projection:** The linear projection creates intermediate vectors $\mathbf{z} \in \mathbb{R}^{\bar{d}}$ for each of the $r$ rank components leading to a memory cost of: $\mathcal{M}_{\text{proj}} = \mathcal{O}(Br\bar{d})$.

2. **Component function computations:** The component MLP $g(\cdot)$ applies a non-linearity to each of the $r \cdot \bar{d}$ projections. If $g$ has a hidden layer of width $h$, standard Autograd must store the hidden states of these sub-networks to differentiate through them. This results in a memory cost of: $\mathcal{M}_g = \mathcal{O}(Br\bar{d}h)$.

3. **Product inputs:** The outputs of the component functions $u_j = 1 + \gamma g(z_j)$ must be stored to compute the gradient of the product operation during backpropagation. This incurs a memory cost of: $\mathcal{M}_{\text{prod}} = \mathcal{O}(Br\bar{d})$.

The total memory requirement for a naive implementation of LRNNs therefore scales as: $\mathcal{M}_{\text{LRNN}} \approx \mathcal{O}(Br\bar{d}h)$. Compared to a standard MLP, the memory footprint is scaled by a factor of roughly $\bar{d} \cdot h$. The primary bottleneck arises during the backward pass of the product term. For a single product $\phi = \prod_{j=1}^{\bar{d}} u_j$, the partial derivative with respect to the $k$-th component is $\frac{\partial \phi}{\partial u_k} = \prod_{j \neq k} u_j$. To compute this without re-evaluation, the expanded tensor of size $B \times r \times \bar{d}$ will be kept in memory in a standard Autograd implementation.

The *effective* memory cost can be significantly reduced by leveraging hardware-aware optimizations that trade computation for memory as outlined below.

**Gradient checkpointing.** The dependence on the component MLP hidden dimension $h$ can be removed via gradient checkpointing. Instead of storing the intermediate states of $g(\cdot)$ (the term $\mathcal{M}_g$), we store only the projected inputs $\mathbf{z}$. During the backward pass, the forward pass of the component MLPs is recomputed on-the-fly. Since $g(\cdot)$ is typically shallow, the computational overhead is minimal, but the memory complexity reduces to: $\mathcal{M}_{\text{LRNN}}^{\text{ckpt}} \approx \mathcal{O}(B \cdot r \cdot \bar{d})$. This removes the multiplier $h$,

which is significant when using more expressive component functions. This is the strategy employed in our experiments when training LRNNs with large projection widths.

**Kernel fusion.** Our standard implementation calculates projections, component functions, and products as separate kernel calls, necessitating repeated reads and writes of the large intermediate tensor of size $B \times r \times \bar{d}$ to global memory. This can be alleviated through kernel fusion, where multiple operations are combined into a single GPU kernel. By loading the projected inputs $\mathbf{z}$, computing $g(\mathbf{z})$, and reducing the product entirely within GPU registers, the intermediate tensors need not be materialized in GPU memory. For the backward pass, the kernel can re-load $\mathbf{z}$, re-compute the values, and generate gradients directly. This effectively shifts the bottleneck from memory capacity to arithmetic intensity.

**Mixed precision and stability.** Finally, we note that the variance-controlled initialization result in Lemma 1 ensures that the product terms do not suffer from numerical instability (exploding or vanishing values). This stability makes LRNNs well-suited for training in half-precision (FP16 or BF16), thereby reducing the activation memory footprint by $50\%$ compared to FP32 training.

## C ABLATION STUDIES

### C.1 ABLATION STUDIES ON PARAMETER-SHARING LRNN MODEL VARIANTS

To investigate the effect of the parameter-sharing LRNN model variants, we perform an ablation study. We compare the performance of the flexible LRNN-SPDER (Flex), with its reduced-complexity variant with shared activation functions (SA) on the INR image representation task for two images. For cameraman (Figure 11 left), we observe that for medium-sized models, the SA variant has better parameter efficiency, achieving higher PSNR at the same complexity as the Flex model. However, at higher parameter counts, the PSNR difference becomes negligible, suggesting a tradeoff between expressivity and efficiency. For retina (Figure 11 right), the SA model has small benefits in parameter efficiency at higher parameter counts. This study indicates while both flexible and reduced-complexity LRNN models perform well, the variant selection is dependent on the task and the computational budget available.

Note that in terms of timing, a typical run of LRNN-SPDER variants with comparable hyperparameters ran approximately 50 iterations per second for Flex and 37 iterations per second for SA.

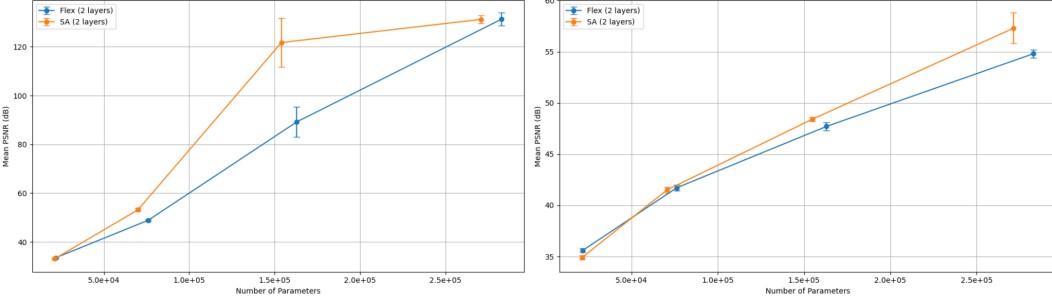

Figure 11: Model complexity neural scaling laws comparing flexible (Flex) and shared activation (SA) variants of LRNN-SPDER on INR image representation tasks for cameraman (left) and retina (right).

### C.2 ABLATION STUDIES ON THE USE OF LAYERNORM

LayerNorm was applied only to LRNN in the numerical experiments and not to the baseline models. The reason for this highlights a key difference in architectural design. The stability of standard SIREN/SPDER models comes from a carefully derived and principled weight initialization scheme that preserves the distribution of activations through the network so that the final output at initialization does not depend on the number of layers. Without this, the accuracy and convergence of deep SIREN/SPDER networks can be very poor.

Our LRNN architecture, however, introduces a fundamentally different activation structure: a product of univariate functions. While we use SIREN-style initialization for each individual component function $g_j$, the statistical properties of their product are distinct and more complex than those of a single sin function. The SIREN initialization scheme, on its own, is no longer sufficient to guarantee that the final output of this product activation will be well-behaved, especially in deep networks. Therefore, we employ LayerNorm as a necessary additional step. It acts as a dynamic normalization layer that explicitly re-centers and re-scales the output of our product-structured activation after it has been computed. This enforces stability by ensuring the inputs to the next layer are consistently well-distributed, regardless of the complex interactions within the product.

To quantify the necessity of this approach, we ran an ablation study on the audio representation task. The result in Table 3 shows that removing LayerNorm significantly degrades performance, resulting in a final error that is orders of magnitude higher. This confirms that normalization is crucial for stabilizing the training dynamics of deep LRNNs by controlling the scale of the activations passed between layers.

Table 3: LayerNorm ablation study on bach audio for LRNN-SPDER.

| LayerNorm | Final MSE after 1000 steps |
|---|---|
| Yes | **3.58e − 5** |
| No | $2.41e - 2$ |

### C.3 Ablation studies on component activations

To provide quantitative evidence on the role of component activations, we ran an ablation study on the Cameraman and Retina tasks. We compared our standard LRNN-SPDER to an LRNN using ReLU and Tanh activations in its component functions. The results in Tables 4a - 4b demonstrate

Table 4: Activation ablation study on Cameraman and Retina images.

(a) Cameraman image.

| LRNN Component Activation | Final PSNR after 1000 steps (dB) |
|---|---|
| SPDER | **107.94** |
| ReLU | 14.40 |
| Tanh | 14.42 |

(b) Retina image.

| LRNN Component Activation | Final PSNR after 1000 steps (dB) |
|---|---|
| SPDER | **47.02** |
| ReLU | 22.55 |
| Tanh | 16.77 |

that using bounded, periodic activations is key for high performance on INR tasks. This highlights a clear design principle for applying LRNNs in practice. Lemma 2 provides further insight. It shows that by learning the parameters of these component functions (e.g., their frequencies), LRNNs can dynamically control their spectral bias, allowing them to represent a much richer set of frequencies than fixed-activation models like SIREN. This is a key reason for their superior performance on complex signals. The most important takeaway is that the LRNN structure itself provides a significant performance boost, regardless of the component activation. Our experiments show LRNN-SIREN outperforms SIREN and LRNN-SPDER outperforms SPDER. This demonstrates the general power of our learnable, factorized activation framework.

## D  Setup for product-structured test function scaling studies

To evaluate the scaling properties of LRNNs on functions well-suited to their architecture, we defined a test function with a sum-of-products structure: $y(\mathbf{x}) = \sum_{\ell=1}^{\tilde{r}} s_\ell \prod_{j=1}^{d} g_j^\ell(x_j)$. Here, we define the component function as $g_j^\ell(x_j) = \sin(n\pi \xi_j^\ell x_j), j = 1, 2, \ldots, d$. For fixed $(n, \tilde{r}, d)$, we randomly sample $s_\ell$ and $\xi_j^\ell$ each from a standard normal distribution. We used Sobol sampling to generate $N_{\text{train}}$ points as the training dataset for both LRNN and MLP architectures, and an independent set of $N_{\text{test}}$ points for evaluating model performance. To generate the scaling laws, $N_{train}$ was fixed at $\sim 3 \times 10^4$ and we trained LRNNs and MLPs with varying number of learnable parameters. As shown in Figure 2 in the main paper, the shallow LRNN (with one hidden layer) architecture achieves better generalization performance at lower model complexity than shallow MLP (with one hidden layer). A decrease in test error with increased model size was not observed for the shallow

MLP, which implies that this model struggled to learn this particular function effectively. The expected trend of decreasing test error with increased model complexity was observed for MLPs when using two hidden layers, with further improvement seen with three hidden layers. However, the shallow LRNN achieved lower test error using fewer parameters than even the three-layer MLP. This indicates that LRNNs exhibit strong scaling capabilities for this class of functions, requiring lower model complexity than standard MLPs to effectively learn data with an inherent sum-of-products structure.

## E  IMAGE REPRESENTATION

The implementation details to reproduce the results for the image representation studies on the cameraman and retina images are presented in Table 5. Note that the author-recommended settings were used for benchmark models, SIREN (Sitzmann et al., 2020) and SPDER (Shah & Sitawarin, 2024).

Table 5: Implementation details for image representation experiments.

| Setting | Details |
| --- | --- |
| Iteration count | All models were trained for 1000 steps. |
| Optimizer | Adam (Kingma & Ba, 2015) was used for all models. |
| Learning rate | $lr = 1 \times 10^{-4}$ for SIREN and SPDER, as recommended in (Sitzmann et al., 2020; Shah & Sitawarin, 2024). $lr = 1 \times 10^{-3}$ for LRNN. |
| Scheduler | No scheduler for SIREN and SPDER, consistent with (Sitzmann et al., 2020; Shah & Sitawarin, 2024). LRNN used StepLR from the standard torch.optim package with step size 100 and decay factor $\gamma = 0.8$ for Cameraman, $\gamma = 0.9$ for color images. |
| Benchmark hyperparameters | Three layers of 256 neurons with $\omega_0 = 30$ for both SIREN and SPDER, per (Sitzmann et al., 2020; Shah & Sitawarin, 2024). |
| LRNN hyperparameters | Two LRNN layers, each of rank 106, $\omega_0 = 30$, $\bar{d} = 16$. Component functions were single-layer neural networks with one hidden neuron and the SPDER activation function $(\sin(x)\sqrt{|x|})$ with $\omega = 30$, yielding a parameter count comparable to the other models. |

Following the encouraging results on the grayscale image, we conducted a similar scaling study on a color image. The aforementioned models were evaluated on the $256 \times 256$ retina image. Figure 12 shows the reconstructed images of all models compared to the ground truth image and the PSNR convergence plot. We observe that LRNN still outperforms the benchmark models, although the margin of improvement is smaller than in the grayscale case.

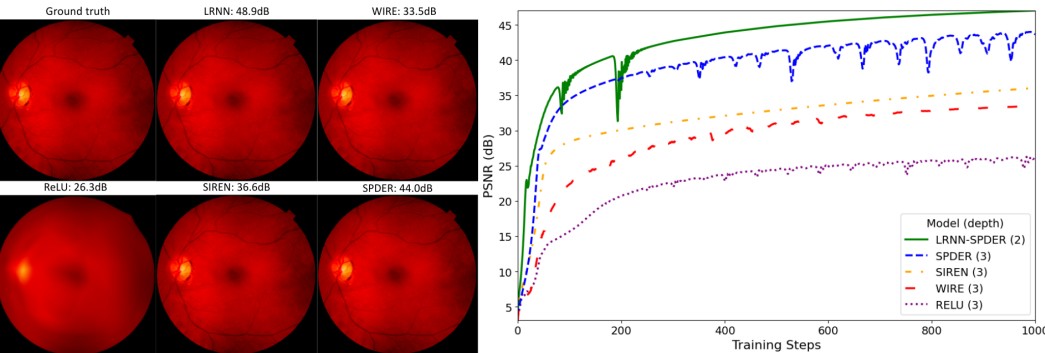

Figure 12: Retina image: ground truth (left), reconstructed images using LRNN, SPDER, SIREN and WIRE models (middle), PSNR convergence history (right).

To evaluate computational efficiency, we measured time-to-solution for specific PSNR targets and presented the results in Tables 6-7. The results for the retina image show that while baseline models may be faster to reach low-quality targets, LRNN achieves the fastest wall-clock time for higher-quality results (over 35dB). This trend is even more pronounced on the cameraman image. While the per-iteration cost of LRNNs can be higher, the time required for high-quality results is significantly better, demonstrating a clear advantage for challenging tasks where quality is paramount.

Table 6: Wall-clock time analysis on cameraman image (seconds).

| Model | Max PSNR | Time to reach | | |
| --- | --- | --- | --- | --- |
| | | 30dB | 35dB | 40dB |
| SIREN | 35.27 | 8.0 | N/A | N/A |
| SPDER | 48.97 | 2.0 | 8.3 | N/A |
| LRNN-SPDER | 107.94 | 5.2 | 6.2 | 8.7 |

Table 7: Wall-clock time analysis on retina image (seconds).

| Model | Max PSNR | Time to reach | | |
| --- | --- | --- | --- | --- |
| | | 30dB | 35dB | 40dB |
| SIREN | 36.04 | 3.5 | 14.1 | N/A |
| SPDER | 43.99 | 2.9 | 4.8 | 12.8 |
| LRNN-SPDER | 47.02 | 5.1 | 5.9 | 9.3 |

We present the iteration time for each model for both the cameraman and retina images in Table 8. It is observed that while LRNN achieves the highest performance in terms of PSNR, it does require more compute time per iteration compared to some benchmarks like SIREN and WIRE (for Retina).

Table 8: Time taken per iteration when training comparably-sized models for INR image representation.

| Image | LRNN | WIRE | SIREN | SPDER |
| --- | --- | --- | --- | --- |
| Cameraman | 0.0453 | 0.0324 | 0.0174 | 0.0332 |
| Retina | 0.0504 | 0.0220 | 0.0072 | 0.0434 |

Studies were performed on higher-resolution images, kodak and parrot, using the code-base from (Saragadam et al., 2023) for the methods we benchmark against. For these experiments, we ran the benchmarks using models with three hidden layers containing 256 neurons. For SIREN and SPDER, we set $\omega_0 = 30$ and for WIRE, we set $\omega_0 = 20, \sigma_0 = 30$, following the respective author's recommendations. The GAUSS and ReLU+PE models from (Saragadam et al., 2023) also used architectures of three hidden layers with 256 neurons each. To match this parameter count, we chose a two-layer LRNN model with each layer being rank 106, $\omega_0 = 30, \bar{d} = 16$, and the component functions being single-layer neural networks with one hidden neuron. As in the previous studies, we used the SPDER activation function $\sin(x)\sqrt{|x|}$ for the MLPs in the LRNNs' product-structured activation function. Each model was trained for 1000 epochs using the Adam optimizer with a batch size of 16384. The LRNN models were trained with a initial learning rate of $10^{-3}$ and a learning rate scheduler (StepLR) from torch.optim with a step size of 100 and a decay rate $\gamma = 0.9$. All other models used the LambdaLR scheduler from the torch.optim package to reduce the learning rate to $0.1\times$ the initial learning rate in the final epoch, as used in the WIRE experiments (Saragadam et al., 2023).

For the higher-resolution kodak image, we compared the LRNN model's reconstructed image and the corresponding gradients at iterations 10 and 500 in Figure 13. We observe a fair reconstruction and gradients at 10, and by step 500, the reconstructed image is virtually indistinguishable. Comparing the training loss and PSNR of LRNN and the 5 benchmarks (SIREN, WIRE, GAUSS,

ReLU (with positional encoding), and SPDER) in Figure 14, we observe that LRNN achieves lower training loss and higher PSNR than the benchmark models throughout training.

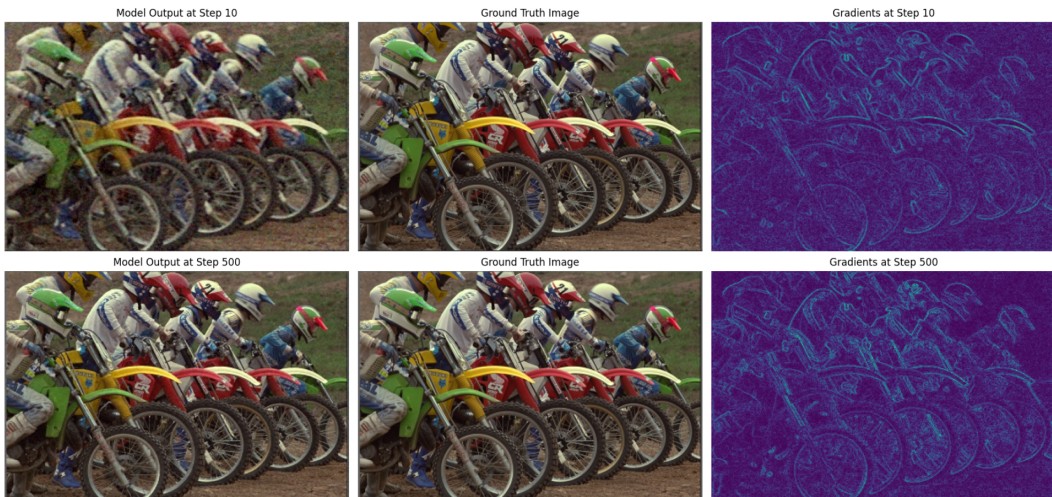

Figure 13: Model output, ground truth and gradient of kodak image at steps 10 and 500.

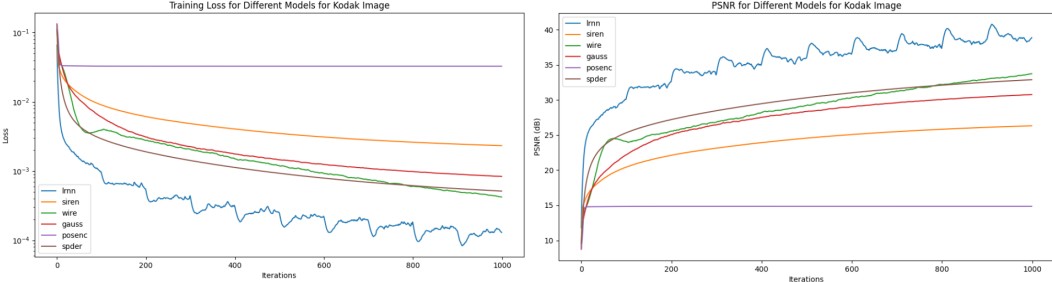

Figure 14: Training loss and PSNR for kodak image over 1000 iterations. The performance of LRNN is compared against SIREN, WIRE, GAUSS, ReLU (with positional encoding), and SPDER.

For the parrot image, we again compare the LRNN model output and gradients at steps 10 and 500 in Figure 15. The model captures the parrot in the foreground well, and significantly detailed by step 500. We observe that the LRNN-reconstructed image at step 500 is virtually indistinguishable from the ground truth. Comparing the training loss and PSNR of LRNN and the 5 benchmarks in Figure 16, we observe that LRNN outperforms the benchmark models throughout training.

### E.1 IMAGENET STUDY

We provide additional analysis of the ImageNet representation study presented in the main paper, where all models were configured with comparable parameter complexity of 200k. Figure 17 illustrates the distribution of PSNR values at multiple training checkpoints. We observe that LRNNs exhibit significantly lower performance variance compared to the baselines. The distributions for SIREN and SPDER show a prevalence of low-PSNR outliers, indicating that these models frequently fail to reach high fidelity. In contrast, LRNNs consistently attain high PSNR values across seeds and images. This indicates that LRNNs offer superior representational capacity and robustness at this parameter budget, consistently satisfying high-precision requirements where baseline architectures often fall short.

Table 9 summarizes the success rates and average wall-clock time required for each model to reach specific PSNR thresholds (33 dB, 35 dB, and 40 dB) across the 1,000 test images. The results highlight that while baseline models like SIREN may have lower per-iteration costs, they struggle to

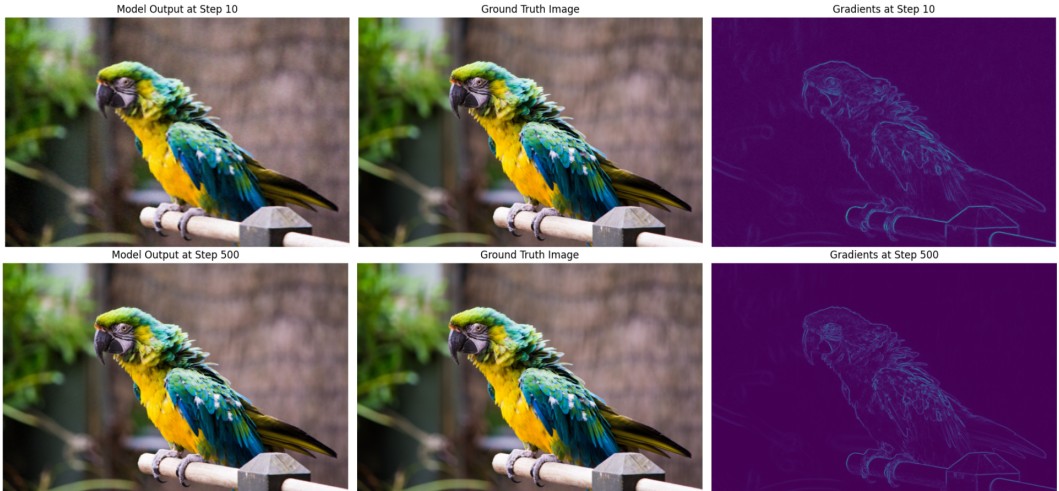

Figure 15: Model output, ground truth and gradient of parrot image at steps 10 and 500.

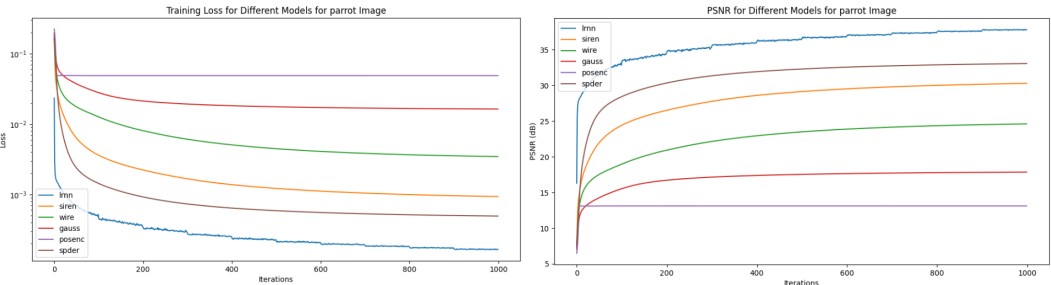

Figure 16: Training loss and PSNR for parrot image over 1000 iterations. The performance of LRNN is compared against SIREN, WIRE, GAUSS, ReLU (with positional encoding), and SPDER.

scale to high precision; for the 40 dB target, LRNN achieves a 100% success rate, whereas SPDER and SIREN succeed in only 26.4% and 1.8% of cases, respectively.

Figure 17: PSNR distribution after different epoch numbers for different models run on 1000 ImageNet images for three seeds each.

Table 9: Time to reach target PSNR for different models on ImageNet (average over 1000 images and 3 seeds). Values show average time $\pm$ standard deviation in seconds, with success rate in parentheses.

| Target PSNR (dB) | Model | Avg Time $\pm$ Std (Success %) | Failures |
|---|---|---|---|
| 33 | LRNN-SPDER | $4.55 \pm 2.87$ (100.0%) | 0 |
| | SPDER | $4.58 \pm 2.29$ (96.7%) | 99 |
| | SIREN | $3.87 \pm 1.23$ (50.7%) | 1480 |
| 35 | LRNN-SPDER | $4.92 \pm 2.90$ (100.0%) | 0 |
| | SPDER | $5.48 \pm 2.47$ (83.8%) | 487 |
| | SIREN | $4.17 \pm 1.28$ (26.5%) | 2204 |
| 40 | LRNN-SPDER | $7.01 \pm 3.27$ (100.0%) | 0 |
| | SPDER | $6.90 \pm 2.11$ (26.4%) | 2207 |
| | SIREN | $4.27 \pm 2.03$ (1.8%) | 2947 |

## F  AUDIO REPRESENTATION

The implementation details for the audio representation study presented in the main paper are outlined here. The code used to produce these results is based on the script provided with (Shah & Sitawarin, 2024) which provides results for baseline models, SIREN (Sitzmann et al., 2020) and SPDER (Shah & Sitawarin, 2024). We create LRNN-SIREN and LRNN-SPDER by using $\sin(x)$ (Sitzmann et al., 2020) and $\sin(x)\arctan(x)$ (Shah & Sitawarin, 2024) activations, respectively, within their MLP-based univariate component functions, comparing them against their baseline counterparts. The benchmark models, SIREN and SPDER have five layers of 256 neurons each. To match this complexity, the LRNN models have three layers, each of rank 118, with $\bar{d} = 10$ and component functions being single-layer neural networks with four hidden neurons. For all models, we set $\omega_0 = 30$ (with inputs normalized to $[-100, +100]$ following (Shah & Sitawarin, 2024) and trained them for 1000 iterations of the Adam (Kingma & Ba, 2015) optimizer with initial learning rate set to $1 \times 10^{-4}$. While SIREN and SPDER had no scheduler on the learning rate to match the original authors' implementation, we chose a StepLR scheduler from the torch.optim package for LRNN with a step size of 100 and $\gamma = 0.8$.

We present the absolute error between the prediction and the ground truth in the time domain and the frequency domain. We observe that for both audio clips (bach in Figure 7 and counting in Figure 18), the LRNN model predictions match the ground truth more closely than the respective

baseline model and hence their error magnitudes are smaller than that of the baselines in both time and frequency domains. This is consistent with the statistical results presented in Table 3 in the main paper. Note that the y-axis for these absolute error frequency domain plots is clipped at 150 for ease of presentation, which excludes a few outliers of higher errors in SIREN and SPDER predictions; despite this, LRNN variants exhibit visibly lower error profiles.

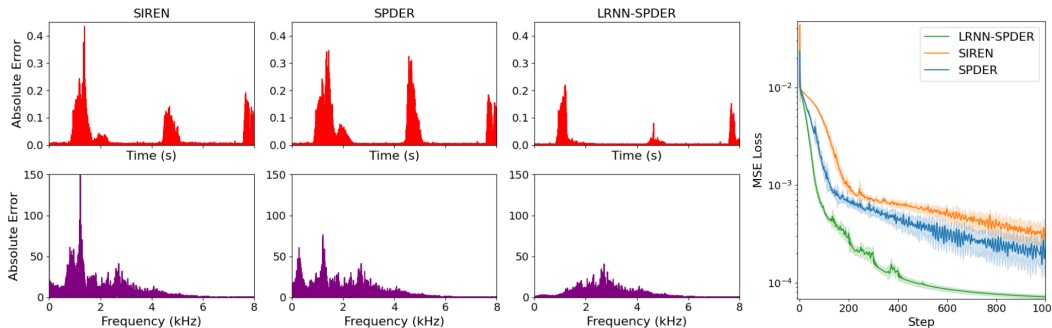

Figure 18: Absolute error in time and frequency domain and convergence of training MSE loss (mean $\pm 1\sigma$) for counting audio representation tasks for comparably sized models.

Table 10: Comparison of MSE loss and $\rho_{AG}$ (mean $\pm$ stddev) across architectures and audio clips at different training steps averaged over 10 runs.

| Metric | Audio | Step | LRNN-SPDER | SIREN | SPDER |
|---|---|---|---|---|---|
| Loss ($\downarrow$) | *bach* | 50 | $\mathbf{1.07 \pm 0.12 \times 10^{-3}}$ | $4.98 \pm 0.44 \times 10^{-3}$ | $2.43 \pm 0.27 \times 10^{-3}$ |
| | | 100 | $\mathbf{2.36 \pm 0.55 \times 10^{-4}}$ | $2.00 \pm 0.18 \times 10^{-3}$ | $1.51 \pm 0.71 \times 10^{-3}$ |
| | | 500 | $\mathbf{1.70 \pm 0.17 \times 10^{-5}}$ | $4.40 \pm 0.39 \times 10^{-4}$ | $3.58 \pm 0.24 \times 10^{-4}$ |
| | | 1000 | $\mathbf{1.01 \pm 0.08 \times 10^{-5}}$ | $1.21 \pm 0.28 \times 10^{-4}$ | $1.12 \pm 0.05 \times 10^{-4}$ |
| | *counting* | 50 | $\mathbf{1.87 \pm 0.26 \times 10^{-3}}$ | $7.24 \pm 0.14 \times 10^{-3}$ | $4.10 \pm 0.34 \times 10^{-3}$ |
| | | 100 | $\mathbf{5.45 \pm 0.45 \times 10^{-4}}$ | $4.27 \pm 0.33 \times 10^{-3}$ | $1.43 \pm 0.22 \times 10^{-3}$ |
| | | 500 | $\mathbf{9.92 \pm 0.82 \times 10^{-5}}$ | $5.72 \pm 0.47 \times 10^{-4}$ | $3.92 \pm 0.75 \times 10^{-4}$ |
| | | 1000 | $\mathbf{7.15 \pm 0.33 \times 10^{-5}}$ | $2.77 \pm 0.56 \times 10^{-4}$ | $2.29 \pm 0.55 \times 10^{-4}$ |
| | *reggae* | 50 | $\mathbf{1.03 \pm 0.35 \times 10^{-2}}$ | $1.57 \pm 0.16 \times 10^{-2}$ | $1.33 \pm 0.19 \times 10^{-2}$ |
| | | 100 | $\mathbf{5.69 \pm 0.67 \times 10^{-3}}$ | $1.37 \pm 0.15 \times 10^{-2}$ | $1.14 \pm 0.38 \times 10^{-2}$ |
| | | 500 | $\mathbf{1.21 \pm 0.14 \times 10^{-3}}$ | $5.02 \pm 0.78 \times 10^{-3}$ | $4.29 \pm 0.51 \times 10^{-3}$ |
| | | 1000 | $\mathbf{7.93 \pm 0.01 \times 10^{-4}}$ | $2.15 \pm 0.63 \times 10^{-3}$ | $2.48 \pm 0.77 \times 10^{-3}$ |
| | *reading* | 50 | $\mathbf{2.72 \pm 0.09 \times 10^{-3}}$ | $3.72 \pm 0.01 \times 10^{-3}$ | $3.09 \pm 0.03 \times 10^{-3}$ |
| | | 100 | $\mathbf{1.98 \pm 0.09 \times 10^{-3}}$ | $3.45 \pm 0.02 \times 10^{-3}$ | $2.58 \pm 0.09 \times 10^{-3}$ |
| | | 500 | $\mathbf{5.63 \pm 0.08 \times 10^{-4}}$ | $1.67 \pm 0.07 \times 10^{-3}$ | $1.41 \pm 0.14 \times 10^{-3}$ |
| | | 1000 | $\mathbf{1.86 \pm 0.03 \times 10^{-4}}$ | $9.98 \pm 0.16 \times 10^{-4}$ | $8.88 \pm 0.25 \times 10^{-4}$ |
| $\rho_{AG}$ ($\uparrow$) | *bach* | 50 | $\mathbf{0.9860 \pm 0.0017}$ | $0.9078 \pm 0.0094$ | $0.9614 \pm 0.0052$ |
| | | 100 | $\mathbf{0.9964 \pm 0.0010}$ | $0.9658 \pm 0.0028$ | $0.9727 \pm 0.0147$ |
| | | 500 | $\mathbf{0.9998 \pm 0.0000}$ | $0.9930 \pm 0.0007$ | $0.9943 \pm 0.0041$ |
| | | 1000 | $\mathbf{0.9999 \pm 0.0000}$ | $0.9986 \pm 0.0005$ | $0.9988 \pm 0.0003$ |
| | *counting* | 50 | $\mathbf{0.9281 \pm 0.0099}$ | $0.5020 \pm 0.0205$ | $0.7989 \pm 0.0254$ |
| | | 100 | $\mathbf{0.9765 \pm 0.0014}$ | $0.7789 \pm 0.0276$ | $0.9425 \pm 0.0115$ |
| | | 500 | $\mathbf{0.9959 \pm 0.0002}$ | $0.9768 \pm 0.0021$ | $0.9861 \pm 0.0014$ |
| | | 1000 | $\mathbf{0.9967 \pm 0.0002}$ | $0.9906 \pm 0.0015$ | $0.9937 \pm 0.0006$ |
| | *reggae* | 50 | $\mathbf{0.7786 \pm 0.0114}$ | $0.6107 \pm 0.0055$ | $0.6896 \pm 0.0055$ |
| | | 100 | $\mathbf{0.8974 \pm 0.0072}$ | $0.6816 \pm 0.0039$ | $0.7475 \pm 0.0090$ |
| | | 500 | $\mathbf{0.9809 \pm 0.0005}$ | $0.9330 \pm 0.0062$ | $0.9449 \pm 0.0035$ |
| | | 1000 | $\mathbf{0.9860 \pm 0.0002}$ | $0.9769 \pm 0.0011$ | $0.9729 \pm 0.0010$ |
| | *reading* | 50 | $\mathbf{0.6315 \pm 0.0171}$ | $0.3100 \pm 0.0075$ | $0.5246 \pm 0.0081$ |
| | | 100 | $\mathbf{0.7430 \pm 0.0133}$ | $0.3883 \pm 0.0070$ | $0.6301 \pm 0.0210$ |
| | | 500 | $\mathbf{0.9508 \pm 0.0082}$ | $0.8069 \pm 0.0070$ | $0.8476 \pm 0.0088$ |
| | | 1000 | $\mathbf{0.9862 \pm 0.0031}$ | $0.9193 \pm 0.0094$ | $0.9324 \pm 0.0104$ |

## G   NUMERICAL SOLUTION OF PDEs

For the task of PDE solution approximation with INR models, we use the Poisson equation $u_{xx} + u_{yy} = f$ on the square domain $\Omega = [-1, 1]^2$, subject to zero Dirichlet boundary conditions from (Liu et al., 2025). The source term is defined as $f(x, y) = -\pi^2(1 + 4y^2)\sin(\pi x)\sin(\pi y^2) + 2\pi\sin(\pi x)\cos(\pi y^2)$ and the exact solution is given by $u(x, y) = \sin(\pi x)\sin(\pi y^2)$. The training objective is composed of two parts: a residual loss over the PDE interior and a boundary loss, formulated as $\mathcal{L}_{\text{PDE}} = \alpha\mathcal{L}_{\text{int}} + \mathcal{L}_{\text{bdry}}$, where

$$\mathcal{L}_{\text{int}} = \frac{1}{n_i}\sum_{i=1}^{n_i} |u_{xx}(z_i) + u_{yy}(z_i) - f(z_i)|^2 \quad \text{and} \quad \mathcal{L}_{\text{bdry}} = \frac{1}{n_b}\sum_{i=1}^{n_b} u^2(z_i).$$

Here, $\{z_i = (x_i, y_i)\}$ are collocation points sampled uniformly within the domain for the interior loss, and on the boundary for the boundary loss. Following (Liu et al., 2025), we set $\alpha = 0.01$.

For the LRNN model, we used a two-layer architecture with $\bar{d} = 12$ and an MLP with one hidden layer with one neuron for each component function together with the SIREN ($\sin(x)$) activation function. SPDER was not considered as a benchmark for this problem as we were unable to achieve competitive performance with the $\sqrt{|x|}$ damping factor. The rank of the LRNN layers were chosen from the set $\{16, 32, 48, 64\}$ to study the scaling properties of LRNNs. For both the LRNN and SIREN models, we chose $\omega = 6.0$ as the frequency parameter for all layers. The MLP used the SiLU activation function following the setting in (Liu et al., 2025). We used a spatial grid of $41 \times 41$ collocation points to compute the loss function. We used a learning rate of $10^{-3}$ for LRNNs and MLPs and a learning rate of $10^{-4}$ for SIREN. All models were trained for 1000 epochs using the Adam optimizer.

Table 11: Comparison of different methods for the two-dimensional Poisson equation (mean squared error). Results for KAN are taken from Liu et al. (2025).

| Method | $n_{params}$ ($\times 10^3$) | $n = 1$ MSE $\downarrow$ | $n = 2$ MSE $\downarrow$ | $n = 4$ MSE $\downarrow$ |
|---|---|---|---|---|
| MLP [2,128,128,128,1] | 34 | $6.4 \times 10^{-6}$ | $1.9 \times 10^{-2}$ | $4.6 \times 10^{-2}$ |
| MLP [2,256,256,256,1] | 132 | $3.6 \times 10^{-6}$ | $1.7 \times 10^{-5}$ | $2.9 \times 10^{-3}$ |
| SIREN [2,128,128,128,1] | 34 | $5.0 \times 10^{-7}$ | $1.2 \times 10^{-4}$ | $1.9 \times 10^{-3}$ |
| SIREN [2,256,256,256,1] | 132 | $1.3 \times 10^{-7}$ | $8.9 \times 10^{-6}$ | $5.5 \times 10^{-4}$ |
| KAN [2,10,1] $G = 10$ | - | $3.6 \times 10^{-5}$ | $1.8 \times 10^{-2}$ | $0.531$ |
| KAN [2,10,1] $G = 20$ | - | $4.9 \times 10^{-2}$ | $6.7 \times 10^{-3}$ | $8.7 \times 10^{-2}$ |
| KAN [2,100,1] $G = 10$ | - | $1.0 \times 10^{-6}$ | $3.6 \times 10^{-5}$ | $9.8 \times 10^{-3}$ |
| KAN [2,100,1] $G = 20$ | - | $0.106$ | $1.8 \times 10^{-2}$ | $8.1 \times 10^{-3}$ |
| KAN [2,10,10,10,1] $G = 10$ | - | $1.4 \times 10^{-4}$ | $1.4 \times 10^{-2}$ | $0.332$ |
| KAN [2,10,10,10,1] $G = 20$ | - | $0.990$ | $0.986$ | $0.964$ |
| **LRNN** [2,16,16,1] | 5 | $8.9 \times 10^{-7}$ | $1.5 \times 10^{-5}$ | $3.7 \times 10^{-3}$ |
| **LRNN** [2,32,32,1] | 16 | $1.5 \times 10^{-7}$ | $\mathbf{2.7 \times 10^{-6}}$ | $\mathbf{5.5 \times 10^{-4}}$ |
| **LRNN** [2,48,48,1] | 34 | $\mathbf{1 \times 10^{-7}}$ | $8.3 \times 10^{-7}$ | $1.5 \times 10^{-4}$ |
| **LRNN** [2,64,64,1] | 57 | $\mathbf{9 \times 10^{-8}}$ | $6.6 \times 10^{-7}$ | $6.6 \times 10^{-5}$ |

The excellent performance of LRNNs on this particular problem (see Table 11 and Figure 8) is consistent with their architectural design, which inherently captures multiplicative interactions. The exact solution possesses a product-separable structure, making it particularly well-suited for approximation by LRNNs. We provide a graphical comparison of results obtained using LRNN, MLP and SIREN models in Figure 19. For LRNNs we used forward mode AD to efficiently compute the Laplacian – this was crucial to prevent memory issues arising from the effective number of intermediate activations associated with its product-structured activation function when the model size is increased.

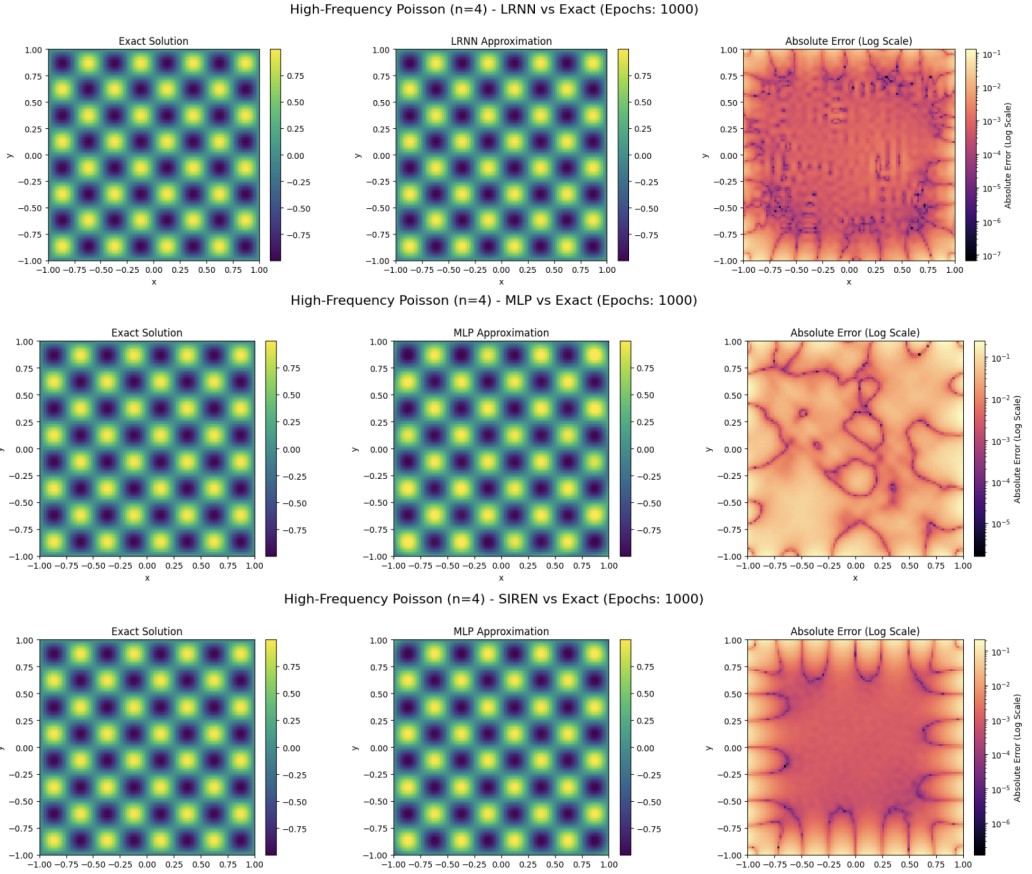

Figure 19: Comparison of approximations obtained using LRNN model with $64k$ parameters and the MLP and SIREN models with $132k$ parameters for the Poisson PDE ($n = 4$). The first row shows the LRNN approximation, the second row shows the MLP approximation, and the third row shows the SIREN approximation. As shown in Table 2 in the main paper, the small LRNN model significantly outperforms the MLP and SIREN models in terms of the MSE.

Note that the results for KAN in Table 11 of the main paper were taken directly from (Liu et al., 2025), as we encountered difficulties in reproducing their reported results using the publicly available code.

## H  CT RECONSTRUCTION ABLATION STUDIES

These experiments were performed by adapting the script for CT reconstruction provided by Saragadam et al. (2023), who introduced this benchmark problem. All models were run for 5000 iterations. The learning rates for the benchmarks were consistent with those recommended by Saragadam et al. (2023): $5 \times 10^{-3}$ for WIRE, $1 \times 10^{-4}$ for ReLU with positional encoding, and $1 \times 10^{-3}$ for SIREN and Gauss. We also chose $lr = 1 \times 10^{-3}$ for LRNN. For the study presented in the main paper and the ablation study presented below for number of projections, the model sizes are as follows: 3 layers of 256 neurons each for all benchmarks, and for LRNN, a two-layer model with each layer being rank 82, $\bar{d} = 10$, and the component functions being single-layer neural networks with four hidden neurons to match the complexity of the benchmarks. For only the LRNN model, a ReduceLRonPlateau scheduler from the torch.optim package was used with a factor of 0.5 and a patience of 25. All other models used the LambdaLR scheduler from the torch.optim package to reduce the learning rate to $0.1\times$ the initial learning rate in the final epoch, as used in the WIRE paper experiments. For the study in the main paper and the ablation study on model complexity, all models were provided with 100 CT projections. The results of the two ablation studies are discussed below.

**Number of CT Projections.**  The CT problem was solved by all five models of similar complexity ($1.8 \times 10^5$ parameters) for different numbers of CT measurements. We compare both the PSNR and SSIM in Table 12. We also consider the reconstructed images in Figure 20 compared to the ground truth in Figure 7 in the main paper. None of the models have a clear reconstruction with only 20 projections, however LRNN and SIREN perform decently well with only 50 projections. As demonstrated in Figure 21, LRNN is the top performer in PSNR at both 50 and 100 projections and in SSIM at 100 and 150 projections. When more projections (150, 200, 300) are available, WIRE achieves the highest PSNR and LRNN is second-best by a small margin of at most 1.53dB. This indicates that while LRNN has some limitations, it is a good choice for this problem, particularly in sparse-view cases with less CT measurements to limit exposure to patients.

Table 12: Comparison of PSNR and SSIM across models for different numbers of CT measurements.

| Number of CT Meas. | LRNN | | SIREN | | WIRE | | Gauss | | ReLU+PE | |
|---|---|---|---|---|---|---|---|---|---|---|
| | PSNR | SSIM | PSNR | SSIM | PSNR | SSIM | PSNR | SSIM | PSNR | SSIM |
| 20 | 25.60 | 0.5449 | 25.59 | 0.5646 | 20.86 | 0.2199 | 22.18 | 0.3351 | **25.75** | **0.5926** |
| 50 | **28.26** | 0.6766 | 27.44 | **0.6817** | 24.63 | 0.4006 | 26.54 | 0.5680 | 26.23 | 0.6180 |
| 100 | **29.13** | **0.7455** | 27.46 | 0.6877 | 28.83 | 0.6413 | 27.84 | 0.6855 | 26.89 | 0.6341 |
| 150 | 29.43 | **0.7557** | 27.45 | 0.6854 | **30.42** | 0.7470 | 27.79 | 0.6945 | 26.97 | 0.6327 |
| 200 | 29.71 | 0.7578 | 27.34 | 0.6874 | **31.24** | **0.7849** | 27.80 | 0.6926 | 26.94 | 0.6221 |
| 300 | 29.77 | 0.7713 | 27.53 | 0.6970 | **30.59** | **0.7739** | 27.99 | 0.7044 | 27.05 | 0.6094 |

**Model Complexity.**  An ablation study was also performed to determine the impact of model complexity, as defined by total number of model parameters, on performance on the CT reconstruction task. All models were provided 100 CT measurements. The range of model size tested varied from $5.9 \times 10^4$ to $19.9 \times 10^4$. Based on Table 13 and Figure 22, we observe that all models with the exception of WIRE do not exhibit any significant difference in PSNR over the range of model complexities–they stay within the same 1dB range. For this reason, the reconstructed images are not presented since they would be difficult to differentiate with the human eye. The models tend to follow a trend of marginal increase in PSNR with significant increase in parameter count. There are some outliers to this trend, most notably the largest WIRE model. This may be explained by the fact that the smaller models had a depth of two, while the largest benchmark models comprised three layers, which may not be well-suited to the WIRE framework. We observe that for 100 projections, LRNN achieves the highest PSNR at all tested model sizes. Overall, this ablation study suggests that model complexity does not have significant impact on performance once a certain minimum complexity is achieved. This is useful to note in problems limited by computational budget.

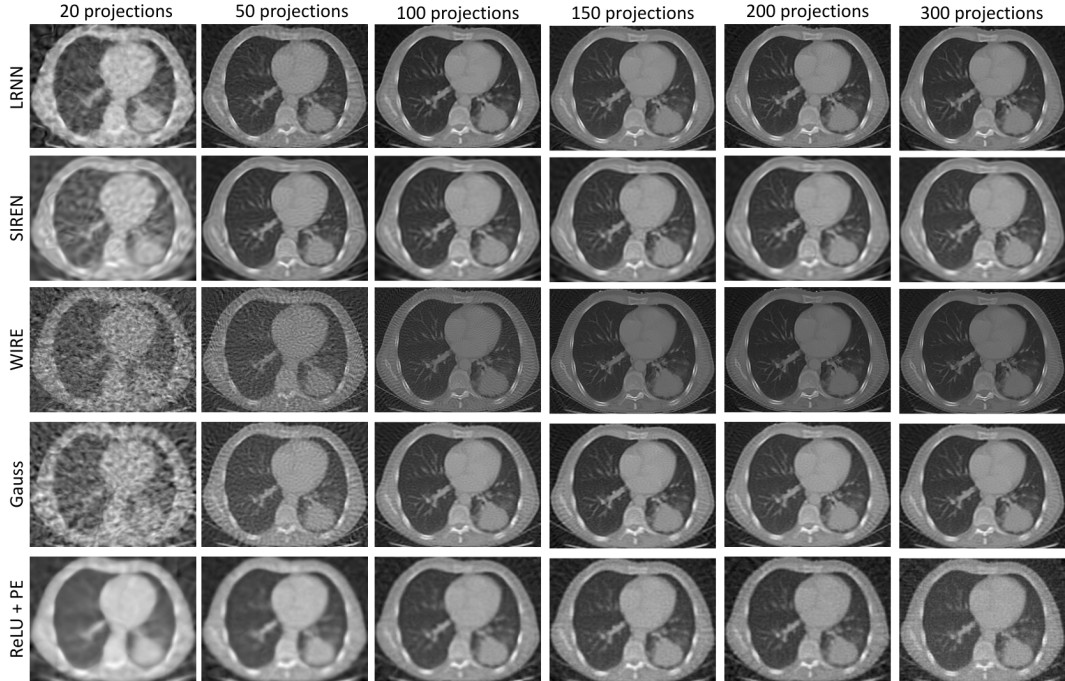

Figure 20: Comparison of CT reconstructed image by similarly complex models for different numbers of projections.

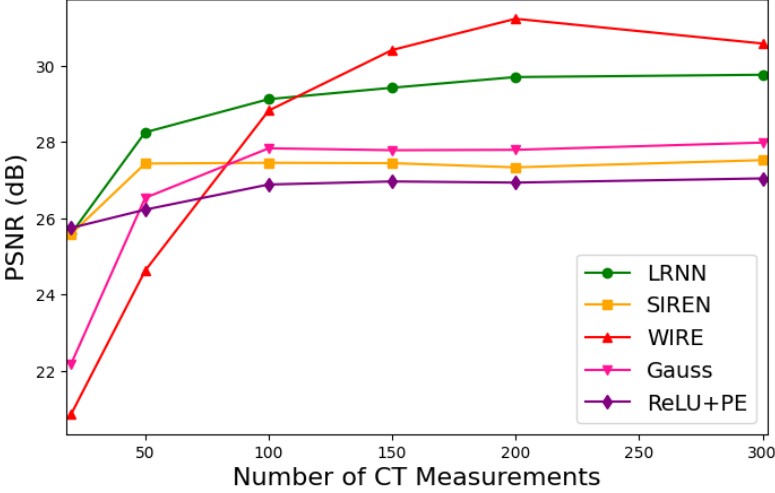

Figure 21: Comparison of PSNR for similarly complex models given different numbers of CT measurements.

Table 13: Comparison of PSNR and SSIM across models for different number of parameters (100 projections).

| Parameters ($\times 10^4$) | LRNN | | SIREN | | WIRE | | Gauss | | ReLU+PE | |
|---|---|---|---|---|---|---|---|---|---|---|
| | PSNR | SSIM | PSNR | SSIM | PSNR | SSIM | PSNR | SSIM | PSNR | SSIM |
| 5.90 | **28.91** | **0.7167** | 26.91 | 0.6530 | 27.88 | 0.6041 | 27.67 | 0.6670 | 26.51 | 0.6142 |
| 9.00 | **29.00** | **0.7297** | 27.09 | 0.6655 | 28.30 | 0.6358 | 27.64 | 0.6727 | 26.48 | 0.6111 |
| 12.85 | **29.63** | **0.7392** | 27.21 | 0.6705 | 28.84 | 0.6479 | 27.73 | 0.6702 | 26.63 | 0.6280 |
| 15.50 | **29.33** | **0.7547** | 27.53 | 0.6907 | 28.93 | 0.6425 | 27.81 | 0.6860 | 26.66 | 0.6269 |
| 18.20 | **29.13** | **0.7455** | 27.46 | 0.6877 | 28.83 | 0.6413 | 27.84 | 0.6855 | 26.89 | 0.6341 |
| 19.90 | **29.43** | **0.7668** | 27.68 | 0.6966 | 26.58 | 0.5005 | 27.28 | 0.5694 | 26.96 | 0.6587 |

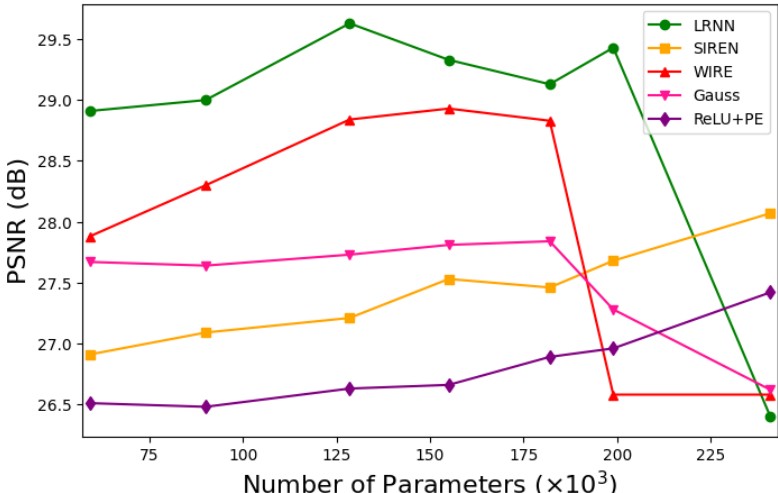

Figure 22: Comparison of PSNR for models of varying complexity given the same numbers of CT measurements.

## I    STUDIES ON IMAGE CLASSIFICATION DATASETS

We present controlled experiments on classification benchmarks to isolate and evaluate the contribution of the LRNN layer as a building block. Our goal is not to achieve state-of-the-art on large-scale datasets—which would require sophisticated architectures and training protocols—but rather to perform rigorous ablation studies comparing LRNN layers directly against their MLP, CNN, and KAN counterparts under identical conditions. We selected MNIST and MNIST-1D datasets for these experiments since they enable controlled comparisons with well-understood baselines and the computational efficiency allows extensive ablation studies. Furthermore, MNIST-1D's procedural generation and shuffle variant specifically test architectural biases.

We consider LRNN models which take the vectorized image as input as well as a ConvLRNN model similar to a standard CNN which involves convolutional and pooling layers followed by an LRNN layer (instead of an MLP block).[4] Our numerical experiments demonstrate that:

1. **MNIST-1D benchmark:** On the MNIST-1D benchmark, both LRNN models consistently outperform their direct MLP and CNN counterparts; see Table 14 and Figure 23.

2. **Shuffled pixels test-case:** On the MNIST-1D benchmark with shuffled pixels, both LRNN models maintain higher accuracy than their counterparts, suggesting they learn different feature representations (see Table 14).

3. **General-Purpose Layer (LRNN vs. MLP vs. KAN):** When treating MNIST as a vector task, our LRNN achieves 98.1% accuracy, which improves upon the standard MLP's 97.0% accuracy and matches the accuracy of a much larger KAN model while being ≈50x faster to train (Table 15).

4. **As a Component in a CNN (ConvLRNN vs. a baseline CNN):** The ConvLRNN model matches the 99.1% accuracy of the baseline CNN on MNIST, but with only one-third of the parameters (77k vs. 225k, Table 15).

We now present the detailed experimental setup and results. We first evaluated LRNN and ConvLRNN on MNIST-1D (Greydanus & Kobak, 2024) against CNN and MLP baselines. All models were run with 5 different random seeds and the average results are presented in Table 14. We observe that ConvLRNN outperforms the CNN in terms of test accuracy on the unshuffled data as well as train and test accuracy on the shuffled data. The CNN has high accuracy on unshuffled data, but struggles

---

[4]Note that it is also possible to modify the LRNN neuron to entirely replace the convolutional layers and incorporate positional encoding instead of typical CNN's fixed encoding. We leave this for future work.

to perform on shuffled data. Although ConvLRNN's performance also degrades on shuffled data, it maintains 98.48% train accuracy (vs CNN's 94.46%) and achieves 64.04% test accuracy—a 7 percentage point improvement over CNN's 57.16%.

Table 14: Classification accuracies on MNIST-1D.

| Model | $n_{params}$ | Unshuffled | | Shuffled | |
|---|---|---|---|---|---|
| | | Train Acc (%) | Test Acc (%) | Train Acc (%) | Test Acc (%) |
| CNN | $5.21 \times 10^3$ | $100.00 \pm 0.00$ | $93.06 \pm 0.80$ | $94.46 \pm 1.48$ | $57.16 \pm 1.29$ |
| MLP | $1.52 \times 10^4$ | $100.00 \pm 0.00$ | $65.00 \pm 1.30$ | $100.00 \pm 0.00$ | $65.20 \pm 1.42$ |
| **ConvLRNN** | $1.79 \times 10^4$ | $100.00 \pm 0.00$ | $\mathbf{94.12 \pm 0.47}$ | $98.48 \pm 1.44$ | $64.04 \pm 2.49$ |
| **LRNN** | $4.63 \times 10^4$ | $100.00 \pm 0.00$ | $67.18 \pm 0.78$ | $100.00 \pm 0.00$ | $\mathbf{67.34 \pm 0.85}$ |

MLPs on the other hand perform consistently whether the data is shuffled or unshuffled, although the unshuffled test accuracy is much lower than that of the convolutional models. LRNN outperforms MLP by 2-3% on both the shuffled and unshuffled data. Overall, ConvLRNN has the highest performance on the unshuffled data while LRNN does the best of these benchmarks on the shuffled data. To graphically illustrate this, we present plots of the train and test accuracy for a typical run in Figure 23. We use this example to demonstrate the versatility of LRNNs and their applicability as a general-purpose layer as illustrated by the broad range of tasks it performs well on.

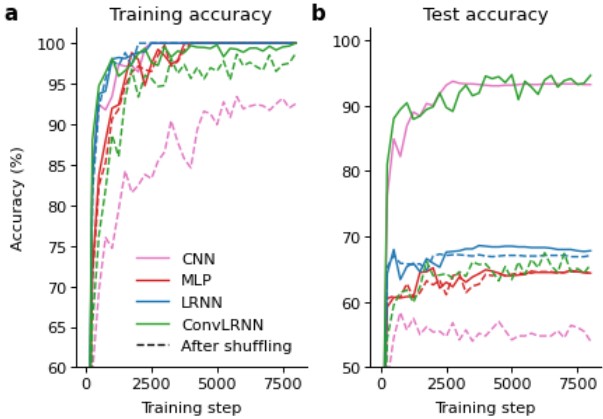

Figure 23: Train and test accuracies on MNIST-1D.

We evaluated five models on standard MNIST: three using the flattened 784-dimensional vector representation (MLP, KAN, LRNN) and two using the 2D image structure (CNN, ConvLRNN). Results are presented in Table 15.

For the vector-based models, LRNN achieves 98.1% test accuracy, outperforming MLP (97.0%) while matching KAN's accuracy. Critically, LRNN trains in 19.38 seconds—comparable to MLP's 16.65 seconds and 55x faster than KAN's 1064 seconds. Despite having fewer parameters than KAN (1.04M vs 1.11M), LRNN delivers equivalent accuracy with dramatically better computational efficiency.

For the convolutional models, ConvLRNN matches CNN's 99.1% accuracy while using only 77k parameters—a 3x reduction from CNN's 225k parameters. This demonstrates that LRNN layers can serve as efficient drop-in replacements in standard architectures, maintaining performance while significantly reducing model complexity.

These controlled experiments demonstrate that the LRNN layer provides tangible benefits—parameter efficiency and computational speed—when used as a drop-in replacement for standard layers. While we do not claim state-of-the-art on classification (our focus remains on continuous signal representation), these results validate LRNNs as versatile building blocks that could

Table 15: Model performance on MNIST.

| Model | $n_{params}$ | Train Loss | Train Accuracy (%) | Test Accuracy (%) | Wall Time(s) |
|---|---|---|---|---|---|
| MLP | $7.95 \times 10^4$ | $9.72 \times 10^{-2}$ | 98.6 | 97.0 | **16.65** |
| KAN | $1.11 \times 10^6$ | $9.24 \times 10^{-3}$ | 100.0 | 98.1 | 1063.99 |
| CNN | $2.25 \times 10^5$ | $\mathbf{7.05 \times 10^{-3}}$ | 99.8 | **99.1** | 382.51 |
| LRNN | $1.04 \times 10^6$ | $1.19 \times 10^{-2}$ | 100.0 | 98.1 | 19.38 |
| ConvLRNN | $\mathbf{7.68 \times 10^4}$ | $8.30 \times 10^{-3}$ | 99.8 | **99.1** | 422.31 |

benefit future architectural designs. Exploring LRNN integration in modern architectures for complex classification tasks remains an interesting direction for future work.

