# OpenReview forum: "Deep Learning with Learnable Product-Structured Activations"
_ICLR.cc/2026/Conference — ICLR 2026 Poster_

### Official Review · Reviewer_yobX · 2025-10-28

**Soundness:** 3
**Presentation:** 3
**Contribution:** 3
**Rating:** 8
**Confidence:** 3

**Summary:**

This paper presents a novel architecture, low-rank separated neural networks (LRNNs), that can be understood as MLPs with learnable activations taking a particular product-factorized form. The authors introduce their architecture, compare it to an MLP, and prove a universal approximation theorem as well as theoretical results studying their architecture's parameter efficiency and inductive bias. They then carry out a thorough empirical study of their architecture, showing that it outperforms prior work in terms of parameter efficiency, representation quality, and convergence rate when used as an implicit neural representation, exhibits good parameter efficiency for solving low-dimensional Poisson equations, and achieves better reconstruction quality on audio representation and CT scan reconstruction.

**Strengths:**

- The paper is well-written, and I was able to follow the main ideas without much trouble. The authors generally do a good job of explaining the significance of their theoretical results. However, I have some questions about the assumptions for Theorem 2, which I detail below.
- I appreciate that the authors have included theoretical analysis for their architecture's parameter efficiency in addition to the standard universal approximation results. I have some questions for Theorem 2, which I detail below.
- The empirical results are strong across the board, though as I note below, LRNN's particularly strong performance in image representation might be overkill for real-world tasks.

**Weaknesses:**

- I'd appreciate the authors providing some more intuition for the assumptions of Theorem 2, since this is a key theoretical result motivating their new architecture. I include some specific questions below.
- The time-to-solution results in Table 1 are a valuable addition to the other data on LRNN-SPIDER's image representation performance. One could argue that the very high PSNRs (>100 dB) reported for LRNN are overkill for real-world tasks, and that the baseline methods reach a threshold of sufficient quality faster than LRNNs. For instance, I cannot discern any visual difference between the SPIDER reconstruction (35.3 dB) and the LRNN reconstruction (107.9 dB) in Figure 3, nor can I discern a visual difference between any of the non-ReLU baselines for the retina image in Figure 10.

**Questions:**

How should we interpret the assumptions for Theorem 2? In particular:

-What are some examples of functions "whose ANOVA decomposition is dominated by terms involving at most $m ≪ d$ variables" and functions that violate this assumption?
- Does the $m_a ≪ d$ condition in Assumption 3 of the proof (lines 1026-1031) correspond to a particular big-O or little-o bound on the interaction order $m_a$?
- How restrictive is Assumption 4? It seems that it must be fairly restrictive, because if $m_a = d$, then one of the terms in the functional ANOVA decomposition involves all $d$ variables and the restrictions on $f$ come entirely from Assumption 4, but the complexity of the LRNN approximation still grows only polynomially with $d$ according to Theorem 2. So Assumption 4 should be pretty strong to mitigate the curse of dimensionality in this way.

---

> ### Author Response · Authors · 2025-11-27
>
> Thank you for your encouraging assessment of our work.
>
> Your questions about the assumptions of Theorem 2 were very helpful in identifying where additional intuition and clarifications were  needed. In the revised manuscript, we have added **Appendix A.6 (Remarks on Theorem 2)**, which provides a discussion of these assumptions together with concrete examples. Below, we address your specific questions and indicate where they are reflected in the revised text.
>
> ## 1. Intuition for Theorem 2
> > “What are some examples of functions ‘whose ANOVA decomposition is dominated by terms involving at most $m$ variables’ and functions that violate this assumption?”
>
> Appendix A.6 now provides representative examples that satisfy or violate the assumptions of Theorem 2. The assumptions characterize functions with **low effective superposition dimension**, in the sense of Caflisch (1997), i.e., only a relatively small number of ANOVA terms are significant, and these terms either involve few variables or admit a low-rank tensor-product structure. Examples that satisfy one or the other of these assumptions include:
>
> * **Pairwise potentials (satisfies Assumption 3):**
>   Functions such as Coulomb or gravitational potentials
>   $V(\mathbf x) = \sum_{i \neq j} \phi(||x_i - x_j||)$
>   are dominated by pairwise interactions ($m_a = 2$). Even when $d$ is large, the number of significant ANOVA terms $N_a$ grows quadratically in $d$, which is compatible with the polynomial bound in Assumption 3.
>
> * **Separable functions (satisfies Assumption 4):**
>   A product state $f(\mathbf x) = \prod_{i=1}^d \sin(x_i)$
>   has full interaction order ($m_a = d$), but it admits a rank-1 representation ($k_{\max} = 1$).
>   Thus, while it does not satisfy the low-interaction-order part of Assumption 3, it satisfies the factorizability condition in Assumption 4, and Theorem 2 still yields polynomial parameter complexity in $d$ in this case.
>
> A canonical example that *violates* the structural assumptions is:
> * **Sign-parity function (violates Assumptions 3 and 4):**
>   $f(\mathbf x) = \prod_{i=1}^d \operatorname{sgn}(x_i)$
>   has full interaction order ($m_a = d$) and high separation rank; it cannot be well approximated by a small sum of smooth product terms. Consequently, the parameter complexity required for LRNNs (or any sum-of-products architecture) to approximate such a function grows exponentially with $d$, and Theorem 2 does not apply.
>
> These examples are now summarized in Appendix A.6 as instances of the function classes covered by Theorem 2.
>
> ## 2. Growth of interaction order (Assumption 3)
> > “Does the $N_a$ condition in Assumption 3 of the proof … correspond to a particular big-O or little-o bound on the interaction order $m_a$?”
>
> Assumption 3 requires that the number of significant ANOVA terms $N_a$ grows at most polynomially with $d$. If the ANOVA decomposition is truncated at maximum interaction order $m_a$, then the number of possible terms behaves as
> $\mathcal{O}(d^{m_a})$.
> Therefore, requiring $N_a = \mathrm{poly}(d)$ implicitly constrains $m_a$ to remain **bounded or slowly growing** with $d$. In particular, if $m_a$ is a fixed constant (e.g., pairwise or triplet interactions), then $N_a$ is polynomial in $d$. If $m_a$ grows linearly with $d$ (for example $m_a \approx d/2$), then $N_a$ becomes exponential and the assumption is violated.
>
> Appendix A.6 now clarifies this connection between the polynomial growth of $N_a$ and the requirement that the effective interaction order $m_a$ remains small relative to $d$.

---

> > ### Author Response · Authors · 2025-11-27
> >
> > ## 3. Restrictiveness of Assumption 4 (factorizability)
> >
> > > “How restrictive is Assumption 4? It seems that it must be fairly restrictive, because if $m_a \approx d$… Assumption 4 should be pretty strong to mitigate the curse of dimensionality in this way.”
> >
> > We agree that Assumption 4 is strong precisely in the regime $m_a \approx d$. Theorem 2 can be viewed as enforcing a *trade-off between interaction order and separation rank*:
> >
> > * If the dominant ANOVA terms involve *few variables* ($m_a \ll d$), then the polynomial bound on $N_a$ from Assumption 3 is sufficient to avoid exponential complexity.
> > * If the dominant terms involve *many or all variables* ($m_a \approx d$), then Assumption 4 requires them to admit a low-rank tensor-product approximation with small $k_{\max}$. In other words, high-order interactions are allowed only when they are “simple” in the sense of having low separation rank.
> >
> > Appendix A.6 describes this as a “conservation of complexity” principle: LRNNs efficiently represent functions that are either **interaction-sparse** (few variables per term, small $m_a$) or **interaction-dense but rank-sparse** (many variables, but small $k_{\max}$). If a function has both global interactions ($m_a \approx d$) and high separation rank, then $k_{\max}$ must grow exponentially with $d$, reintroducing the curse of dimensionality; Theorem 2 does not claim to avoid this.
> >
> > In many scientific and signal-processing applications, models fall into one of the two structured regimes highlighted above (for example, pairwise potentials or separable/multiplicative structures), which is why we view these assumptions as realistic for the target use cases of LRNNs.
> >
> > ## 4. “Overkill” PSNR and visual quality
> >
> > > “One could argue that the very high PSNRs (>100 dB) reported for LRNN are overkill for real-world tasks … I cannot discern any visual difference…”
> >
> > We agree that PSNR levels above approximately 40 dB are visually indistinguishable for typical images. However, the potential for this level of fidelity is relevant for two reasons:
> >
> > - For numerical solution of PDEs, we need high precision to ensure the derivatives (e.g., the Laplacian) of the approximation to the field variable satisfy the governing equation. A small error in the field variable approximation can lead to large errors in the derivatives, impacting physical validity of the final solution.
> >
> > - The ability to reach 100 dB (while baselines plateau at 40-50 dB) serves as a stress test for the architecture's learning capacity. It demonstrates that LRNNs are parameter-efficient and do not suffer from the "spectral bias" saturation that limits standard neurons that additively compose features, confirming the theoretical claim of superior expressivity. The new large-scale numerical study we include in Section 4 on 1,000 images from ImageNet (see Figure 4, Figure 17, and Table 9) provides  additional numerical evidence supporting this point.
> >
> >
> > To reflect your comment, we now explicitly note in Section 4 that PSNRs above the usual perceptual threshold should be interpreted as a diagnostic of representational capacity rather than as a practically necessary target for image quality.
> >
> > ---
> >
> > Once again, we thank you for the thoughtful questions and detailed reading. Your comments have led to a clearer exposition of Theorem 2 and its assumptions, and to improved contextualization of the image-representation results.

---

### Official Review · Reviewer_XNN7 · 2025-11-01

**Soundness:** 2
**Presentation:** 2
**Contribution:** 2
**Rating:** 4
**Confidence:** 4

**Summary:**

This paper presents a novel approach to expressing a neural representation, primarily utilizing MLPs, by introducing a low-rank formulation and incorporating activation functions as learnable univariate functions. The paper presents preliminary results on the tasks of signal representation and inverse problems, including sparse-CT and super-resolution.

**Strengths:**

- The premise of decomposing each layer into a low-rank layer is interesting and the preliminary results indicate its promising ability for signal representation and inverse problems.

- The paper presents a careful explanation of how LRNNs are implemented, along with added derivations to ensure convergence and stability.

- The paper presents promising initial results, highlighting the impressive ability of LRNNs for signal representation and PDEs.

**Weaknesses:**

Lacks rigorous evaluation: The paper doesn’t show results on image representation or audio representation across a wide variety of samples. The ability of LRNNs to perform across a wide variety of data, not just a few select images, is crucial to understanding the value of this contribution.  Please display results on a large number of examples (e.g., 1000) such as Celeb-A, FFHQ, or ImageNet. Previous papers on signal representations (Functa[1,2], TransINR[3], STRAINER[4]) have shown a rigorous assessment of INRs on the task of image fitting. Furthermore, for tasks such as super-resolution and CT reconstruction, only one image/signal is used, which does not accurately convey the general performance of LRNNs.

Insufficient analysis of LRNN: The paper does not address why a deep low-rank network is able to correctly capture high-quality image or audio-level details. Typically, low-rank approximations work when a distribution can be easily marginalized into “r” independent vectors. Given that LRNNs use a small “r”,  this seems rather counterintuitive as to why it would work. Any qualitative or formal insights would greatly benefit the paper.

[1]From data to functa: Your data point is a function and you can treat it like one, Dupont et.al
[2]Spatial Functa: Scaling Functa to ImageNet Classification and Generation, Bauer et.al.
[3]Transformers as Meta-Learners for Implicit Neural Representations, Chen et.al
[4]Learning Transferable Features for Implicit Neural Representations, Vyas et.al.

**Questions:**

1. For tasks such as 3D scene reconstruction (NERFs), where signal representations are most widely used, have the authors tried/observed any significant performance improvements (faster convergence?)

2. L78: The Authors claim that a low-rank assumption would enhance expressivity. However, that relies on a few key assumptions: for the data to have a low rank of “r”, it would mean that it can be marginalized into r independent variables/distributions. Only in such a case could the LRNN’s design “enhance expressivity.” Is this true in the case of natural images or audio’s frequency spectra?

3. Compared to PCA with “r” components, what benefit does a 2-layer LRNN yield, besides it being nonlinear?

4. The results for the cameraman image appear to be inconsistent with those in Table 7 and Figure 3 (PSNR plot). In Table 7, it is reported that SPDR reaches 30 dB in 2 seconds, before LRNNs. However, the plot shown in Figure 3 doesn’t concur with this finding. It seems that both the figure and the table correspond to the same experiment. Please explain the discrepancy.

---

> ### Author Response · Authors · 2025-11-27
>
> Thank you for the careful assessment. Your comments on evaluation and low-rank expressivity have been very helpful in strengthening the paper. We have uploaded a revised version that addresses your feedback.
>
> ## 1. Evaluation rigor
> > "Lacks rigorous evaluation: The paper doesn’t show results on image representation or audio representation across a wide variety of samples. The ability of LRNNs to perform across a wide variety of data, not just a few select images, is crucial to understanding the value of this contribution. Please display results on a large number of examples (e.g., 1000) such as Celeb-A, FFHQ, or ImageNet. "
>
> We agree that robustness across a large and diverse dataset is important to validate the utility of the architecture. In the revised manuscript, we now include a large-scale study on 1,000 ImageNet images (with 3 random seeds per image, totaling 3,000 trained models for each architecture); please see **Section 4** and **Appendix E.1**.
>
> We evaluated the architectures at a comparable parameter count (~200k) based on the wall-clock time required to reach distinct target PSNR levels (33 dB, 35 dB, 40 dB). As shown in the new Figure 4, Figure 17, and Table 9, LRNNs consistently outperform baselines in high-fidelity reconstruction. We highlight three key findings from this study:
>  * **Success rate:** LRNN-SPDER attains 40 dB on 100% of images. In contrast, SIREN fails to reach this threshold in 98.2% of cases, and SPDER fails in 73.6% of cases.
> * **Robustness (low variance):** LRNNs exhibit substantially lower variance in final PSNR across random seeds compared to baselines (see Figure 17).
> * **Time-to-solution:** While LRNNs incur a higher computational cost per iteration, they require substantially fewer epochs to converge. Consequently, for high-precision targets, LRNNs actually achieve a faster time-to-solution than the computationally cheaper baselines.
>
> We believe this extensive evaluation confirms that LRNNs are a robust and general-purpose architecture, addressing the concern regarding evaluation rigor.
>
> In addition to the large-scale image study, we ensured diversity in the audio benchmarking studies by selecting four distinct signal types: instrumental music, male speech, female speech, and music with vocals. This confirms that the architectural advantages hold across different signal modalities and spectral characteristics.
>
> **Scope of evaluation:** The Functa, Spatial Functa, TransINR, STRAINER architectures focus on meta-learning or learning priors over a dataset of signals. Our contribution is at the level of the per-signal architecture, in the same spirit as SIREN or WIRE. The new ImageNet study shows that LRNNs perform robustly across a wide range of images and therefore provide a strong building block that could be combined with these meta-learning approaches in future work.
>
> ## 2. Low-rank and expressivity
> > "L78: The Authors claim that a low-rank assumption would enhance expressivity. However... for the data to have a low rank of 'r', it would mean that it can be marginalized into r independent variables/distributions... Is this true in the case of natural images or audio’s frequency spectra?"
>
> This is an important point, and we appreciate the opportunity to clarify terminology. The reviewer is correct that a single rank-1 term implies separability (independence). However, the "enhanced expressivity" claim on L78 refers to specific properties of the LRNN architecture compared to standard MLPs.
>
> - *Sum-of-products vs independence.* Shallow LRNNs represent functions as a *sum* of rank-1 terms, i.e., $f(x) \approx \sum_{\ell=1}^r s_\ell \prod_{j=1}^\bar{d} (1+\gamma g_j^\ell (z_j^\ell))$, and are capable of universal approximation (Theorem 1). Theorem 2 shows that LRNNs can mitigate the curse of dimensionality if the target function has low effective separation rank. The analysis does not make assumptions on the data distribution. Please see Appendix A.6 in the revised paper, which discusses Theorem 2 in detail.
> - *Multiplicative inductive bias:*  A standard MLP neuron additively composes features, making it difficult to represent multiplicative interactions without significant depth. An LRNN neuron explicitly captures such interactions via product-structured activations. In this sense, “enhanced expressivity” refers to the *parameter-efficiency* with which LRNNs capture high-order interactions, relative to standard MLP neurons.
> - *Representation of natural signals:* Natural signals are often structurally separable in a local basis (e.g., a Gabor filter which is the product of a Gaussian and a sinusoid). LRNNs learn such univariate “building blocks” and combine them multiplicatively, which aligns well with these generative structures.
>
> ## Revision made
> Prompted by your observation, we have revised this paragraph on Page 2 to clarify that we leverage "low-rank function decomposition" to capture high-order interactions, rather than as a technique for model compression.

---

> > ### Author Response · Authors · 2025-11-27
> >
> > ## 3. Comparison to PCA with (r) components
> >
> > > “Compared to PCA with ‘r’ components, what benefit does a 2-layer LRNN yield, besides it being nonlinear?”
> >
> > We wish to clarify the key differences in the settings for PCA and LRNNs. PCA computes a low-rank approximation of a matrix, i.e., it learns a low-dimensional linear subspace for a set of observed vectors. In contrast,  LRNNs learn a deep low-rank decomposition of a multivariate function (approximating a continuous signal as a sum of separable product terms) given a set of sample observations of the target function. As noted in Section 2, the SVD/PCA approach does not have a unique generalization to tensors of order $>2$. To clarify, the "rank" in LRNN refers to the *separation rank* (how many separable product terms are needed to approximate the function), which is distinct from the *matrix rank* used in PCA.
> >
> > If the question implies contrasting LRNNs to a method that first compresses the input features using PCA and then fits a model in that subspace (as in principal component regression), the specific benefits of LRNNs are:
> >
> > - **Unsupervised vs. supervised feature extraction:** PCA identifies directions of maximum global variance, independent of the prediction task, and may discard low-variance directions that are critical for approximating the target function. LRNNs learn the component functions $g_j$ based on the prediction error, allowing them to capture relevant features regardless of their input variance.
> >
> > - **Multiplicative inductive bias:** A PCA-based representation relies on the additive superposition of features. This structure cannot easily capture interactions between variables (e.g., $x_1$ modulating $x_2$) without explicit feature engineering. LRNNs are built on multiplicative composition, enabling them to efficiently discover and represent high-order multiplicative interactions that a linear manifold representation cannot capture.
> >
> > ## 4. Cameraman Test Case (Table 7 vs. Figure 3):
> >
> > > “The results for the cameraman image appear to be inconsistent with those in Table 7 and Figure 3… Please explain the discrepancy.”
> >
> > Thank you for drawing attention to this. The apparent discrepancy arises because the two results are plotted against different independent variables:
> >
> > * **Figure 3** shows **PSNR versus training iterations**.
> > * **Table 6** (formerly Table 7) reports **PSNR versus wall-clock time**.
> >
> > Because LRNNs have a higher per-iteration cost than SPDER, SPDER can take more optimization steps per second and therefore reaches a modest PSNR level (30 dB) faster in wall time, even though its per-step progress is smaller. LRNNs, in contrast, continue to improve and ultimately reach much higher PSNR (e.g., 107 dB versus 49 dB for SPDER) that the baseline does not attain regardless of training time.
> >
> > To prevent confusion, we clarify in the caption of Figure 3 that the x-axis is “Number of training iterations,” and we explicitly state in the text that Table 6 (formerly Table 7) reports wall-clock time.
> >
> > ## 5. NeRFs and 3D Reconstruction:
> > > “For tasks such as 3D scene reconstruction (NeRFs)… have the authors tried/observed any significant performance improvements (faster convergence)?”
> >
> > We have not yet integrated LRNNs into a full NeRF pipeline, which involves additional computational and algorithmic components (e.g., volumetric rendering, sampling strategies) beyond the representation network itself. Given the highly encouraging  results on image, CT reconstruction, and PDE benchmarks, we regard 3D scene reconstruction as a natural and promising extension where the multiplicative structure of LRNNs could be particularly beneficial for modeling view-dependent effects and fine detail. We have added this as a direction for future work in the conclusion of the revised paper.
> >
> > ---
> >
> > We hope that the new large-scale ImageNet evaluation, together with the clarified discussion of low-rank function decompositions and multiplicative expressivity, addresses your concerns regarding both the empirical rigor and the theoretical soundness of the proposed architecture.  We thank you for your feedback that motivated these improvements.

---

### Official Review · Reviewer_VXpv · 2025-11-01

**Soundness:** 3
**Presentation:** 3
**Contribution:** 3
**Rating:** 6
**Confidence:** 3

**Summary:**

The paper introduces Deep Low-Rank Separated Neural Networks (LRNNs) — a new neural network architecture that generalizes multilayer perceptrons (MLPs) by endowing each neuron with a learnable, product-structured activation function. Instead of relying on fixed non-linearities (e.g., ReLU, Tanh, Sine), LRNNs construct activations as multiplicative compositions of learnable univariate functions. This design captures multiplicative and higher-order interactions efficiently, inspired by low-rank separated representations and tensor decomposition theory. The authors provide theoretical guarantees, including universal approximation, variance-controlled initialization, and curse of dimensionality mitigation for low-rank structured functions. Empirical results are demonstrated on images, audio, PDE solving, sparse-view CT reconstruction. Overall, LRNNs offer a unified, theoretically grounded architecture that merges ideas from tensor decompositions, implicit neural representations (INRs), and adaptive activations.

**Strengths:**

1. The paper is very well-written and motivated.
2. Related work section is well rounded.
3. The paper is easy to follow.
4. The idea of learnable product-structured activations is novel unifying classical low-rank function decomposition with modern deep learning architectures.
5. The theoretical results are are well motivated and rigorously derived.
6. Experiments span multiple domains such as image and audio INRs, PDE solving, and CT reconstruction — demonstrating the versatility of LRNNs.
7. LayerNorm regularization, parameter sharing across layers, and use of small MLPs for component functions are well justified.
8. I believe this can open up new architectural choices for many domain including scientific machine learning.

**Weaknesses:**

1. Explicit computation requirement comparison is missing for the numerical experiments, these would make the proposed architecture much more favourable.
2. The paper acknowledges that LRNNs require higher memory due to product-term storage. More quantitative profiling (time per iteration, memory vs MLPs) would help assess practicality for large-scale settings.
3. The plot(s) can be made more clearly legible.
4. Some ablations regarding the architecture choices should be discussed in the main paper.

**Questions:**

1. Can author(s) discuss how the effective separation rank evolve during training? Does over-parameterizing  r lead to redundancy or implicit sparsity (similar to low-rank dropout)?
2. The variance-controlled initialization ensures bounded gradients theoretically. In practice, do we observe gradient explosion/vanishing for deeper networks?
3. Can there be a discussion around the interpretability of LRNNs?
4. Could the product-structured activation principle be incorporated into attention or convolutional blocks to yield hybrid LRNN-transformer models?
5. Given the increased per-layer product cost, can the author(s) comment on the potential use of mixed-precision or kernel-fusion optimizations to reduce compute overhead?

---

> ### Author Response · Authors · 2025-11-27
>
> Thank you for the careful evaluation and for highlighting both the novelty of LRNNs and the rigor of the theoretical and empirical components. We particularly appreciate your comments on computational cost, memory, and practical deployment, which we address below. All changes referenced here are reflected in the revised manuscript.
>
> ## 1. Computational cost & time-to-solution
> > "Explicit computation requirement comparison is missing for the numerical experiments, these would make the proposed architecture much more favorable."
>
> We agree that time-to-solution is crucial for assessing practicality. In the revised manuscript, Section 4 and Appendix E.1 now include a large-scale study on 1,000 ImageNet images (3 random seeds per image leading to a total of 3,000 trained models for each architecture). We measure the wall-clock time required by each architecture with comparable parameter count of ~200k to reach target PSNR levels (33 dB, 35 dB, 40 dB).
>
> We summarize below our key observations (Section 4, Figure 4; Appendix E.1, Figure 17, Table 9):
>
> - **Higher per-iteration cost but faster convergence in practice:** LRNN-SPDER incurs a higher per-step cost than SIREN/SPDER (due to the product-structured activation), but requires substantially fewer epochs to reach a given target PSNR. For high-precision targets, this leads to favorable time-to-solution.
> - **High-fidelity regime (40 dB):** LRNN-SPDER reaches 40 dB in 100% of runs (average 7.01 seconds). In contrast, SIREN fails to reach 40 dB in 98.2% of runs, and SPDER fails in 73.6% of runs.
> - **Robustness:** As shown in Appendix E.1 (Figure 17 and Table 9), LRNN-SPDER exhibits substantially lower variance in final PSNR compared with SIREN and SPDER, which frequently plateau at lower PSNR.
>
> Overall, these results indicate that although LRNNs are somewhat more expensive per iteration, their robustness and convergence properties make them competitive or favorable in time-to-solution when high-fidelity reconstructions are required.
>
> ## 2. Memory usage & optimization
> > "The paper acknowledges that LRNNs require higher memory due to product-term storage. More quantitative profiling (time per iteration, memory vs MLPs) would help assess practicality for large-scale settings. Given the increased per-layer product cost,  can the author(s) comment on potential use of mixed-precision or kernel-fusion?"
>
> We fully agree that memory is an important practical consideration. The revised manuscript includes a new **Appendix B.2: Memory Complexity Analysis and Optimization Strategies**. Appendix B.2 provides:
>
> * A *scaling analysis* showing that a naive LRNN implementation has activation memory $M_{\text{LRNN}} \approx O(Br\bar{d}h)$,
>   compared with $O(B r)$ for an MLP, where $B$ is batch size, $r$ is separation rank, $\bar{d}$ is projection width, and $h$ is the hidden width of the component MLPs.
>
> * *Gradient checkpointing*, which eliminates the dependence on $h$ by recomputing the shallow component MLPs during backpropagation. This is the strategy used in our large-$\bar{d}$ experiments.
>
> * *Kernel fusion*, where projection, component evaluation, and product reduction are fused into a single GPU kernel so that the intermediate $B \times r \times \bar{d}$ tensor need not be fully materialized in memory. This shifts the bottleneck from memory to arithmetic intensity.
>
> * *Mixed precision*, enabled by the variance-controlled initialization in Lemma 1. Since the product activations are stabilized by the scaling $(\gamma = \bar{d}^{-1/2})$, they do not suffer from numerical instability, making LRNNs well-suited for mixed-precision training.
>
> We believe these implementation strategies address the reviewer’s concern about practicality in memory-constrained or large-scale settings.

---

> ### Author Response · Authors · 2025-11-27
>
> ## 3. Effective separation rank and implicit sparsity
> > “Can the authors discuss how the effective separation rank evolves during training? Does over-parameterizing $r$ lead to redundancy or implicit sparsity (similar to low-rank dropout)?”
>
> We have added a discussion in **Appendix A.2** (Interpretability and Interaction Analysis). LRNNs provide two key pathways for controlling model capacity: projection width $\bar{d}$ and separation rank $r$.
>
> * **Interaction order and ARD via $\bar{d}$:** The product structure $\prod_{j=1}^{\bar{d}} (1 + \gamma g_j(z_j))$ admits a natural mechanism for interaction pruning. If a projected coordinate $z_j$ is irrelevant, the corresponding component function tends toward $g_j \approx 0$, so the factor $1 + \gamma g_j(z_j)$ approaches 1 and the dimension effectively drops out of the product.
>
> * **Separation rank $r$:** In our experiments, over-parameterizing $r$ typically leads to a *distributed representation* rather than hard sparsity in rank. The additional rank components provide redundancy that eases optimization and improves robustness, similar in spirit to over-parameterization in standard deep networks.
>
> At the same time, LRNNs offer a clear path to *explicit rank sparsification*: using a sparsity inducing regularizer on the rank-specific scale vectors $s_\ell$ can zero out entire rank-1 components. We did not implement this in the current work, but the structure of LRNNs makes it straightforward to add in future studies.
>
> In summary, over-parameterizing $r$ primarily yields redundancy that aids optimization, whereas over-parameterizing $\bar{d}$ combined with weight decay induces an implicit bias toward lower effective interaction order.
>
> ## 4. Interpretability
> > "Can there be a discussion around the interpretability of LRNNs?"
>
> Thank you for your suggestion. We have added **Appendix A.2: Interpretability and Interaction Analysis**. There we emphasize that LRNNs are similar in spirit to generalized additive models with interactions:
>
> * Each neuron computes $\varphi_{\ell}({\bf z}^{\ell}) = \prod_{j=1}^{\bar{d}} (1+\gamma \, g_j^{\ell}(z_j^{\ell}))$, where each $g_j^{\ell} : \mathbb{R} \to \mathbb{R}$ is univariate and can be visualized directly.
>
> * By plotting the learned LRNN component functions, $g_j^{\ell}$, one can inspect the nonlinear transformation applied to each projected feature (e.g., linear trend, threshold, or periodic modulation).
>
> * By computing empirical variances $\mathrm{Var}[g_j^{\ell}]$ over a validation set, we can construct “interaction heatmaps” that indicate which coordinates are active for each neuron. For example, if a neuron shows high variance only for components ($j = 1$) and ($j = 4$), it is effectively modeling the interaction between those dimensions.
>
> This level of structured interpretability is difficult to obtain from standard MLPs, where interactions are entangled across dense layers.
>
> ## 5. Hybrid Architectures
> > "Could the product-structured activation... be incorporated into attention or convolutional blocks?"
>
> Yes, we believe this is a fruitful direction. We present a study in Appendix I, where LRNN blocks replace the standard MLP layers in a CNN. This ConvLRNN matches the accuracy of the baseline CNN with ~3x fewer parameters (77k vs. 225k), indicating that LRNNs can yield substantial parameter efficiency without degrading performance. Beyond replacing dense layers, we envision using LRNNs in a sliding-window fashion to define nonlinear receptive fields. Unlike standard convolutions followed by pointwise nonlinearities, such LRNN-based local blocks could model higher-order interactions within the receptive field directly, potentially reducing the depth required for effective feature extraction.
>
> Replacing MLP sublayers in Transformers with LRNN blocks is also a promising direction. In principle, LRNNs can enable more parameter-efficient modeling of high-order interactions between token features than standard MLPs. In summary, we view the present work as establishing LRNNs as a versatile building block that can be integrated into other architectures.
>
> ## 6. Gradient stability in deep LRNNs
> > “The variance-controlled initialization ensures bounded gradients theoretically. In practice, do we observe gradient explosion/vanishing for deeper networks?”
>
> Empirically, we do not observe gradient explosion or vanishing when combining our variance-controlled initialization with LayerNorm. Section 3.1 and Appendix A.3 (Lemma 3 and Corollary 1) show that with $\gamma = \bar{d}^{-1/2}$, the activation variance and the sum of gradient variances across coordinates remains bounded, independently of $\bar{d}$.
>
> Appendix C.2 presents an ablation study showing that removing LayerNorm leads to degradation in accuracy, confirming that normalization is critical for stable deep LRNN training. With LayerNorm and the recommended initialization, we did not encounter numerical instability in any of our experiments, including deeper architectures.

---

> > ### Author Response · Authors · 2025-11-27
> >
> > ## 7. Architecture choices and ablations
> > > "Some ablations regarding the architecture choices should be discussed in the main paper."
> >
> > Thank you for this suggestion. We have included a new paragraph in Section 4 summarizing the insights from the ablation studies:
> >
> > > Prior to presenting task-specific results, we summarize key architectural insights gained through
> > extensive ablation studies (detailed in Appendices C–H): (i) Stability: The multiplicative structure
> > of LRNNs alters activation statistics compared to additive networks, making LayerNorm essential
> > for convergence (Appendix C.2); (ii) Component Selection: Using periodic activations (e.g., SIREN,
> > SPDER) within the univariate components is crucial for minimizing spectral bias in high-frequency
> > tasks (Appendix C.3); and (iii) Robustness: LRNNs provide excellent performance in the sparse-
> > data regime, maintaining high reconstruction fidelity (Appendix H).
> >
> > We found that the performance of LRNNs is not sensitive to the depth of $g$. In fact, keeping $g$ simple (shallow/narrow) and allocating the parameter budget to increasing the separation rank ($r$) or projection width ($\bar{d}$) is the most effective scaling strategy.
> >
> > As shown in our scaling law experiments (Figure 2), performance scales reliably with total parameter count. Since $g$ is small, the parameter count is dominated by the linear projections. This aligns with our theoretical analysis (Theorem 2): the approximation power for low-rank functions is driven by the number of product terms ($r$) and the dimensionality of the subspace ($\bar{d}$), rather than the complexity of the individual factors. This validates the core premise of LRNNs: achieving complexity through the interaction of simple components rather than the depth of components.
> >
> > ## **Summary of revisions**
> >
> > - Added new large-scale study on ImageNet time-to-solution and success rates with further details and figures in Section 4 and Appendix E.1.
> > - Added Appendix A.2 with a discussion on interpretability.
> > - Added Appendix B.2 detailing memory complexity, kernel fusion, and mixed-precision suitability.
> > - Updated plots for improved legibility.
> >
> > ---
> >
> > Thank you again for your thoughtful questions and suggestions. They have led to a clearer presentation of LRNNs’ computational profile, interpretability, and architectural trade-offs, as well as additional empirical evidence on time-to-solution.

---

### Official Review · Reviewer_XhCn · 2025-11-02

**Soundness:** 3
**Presentation:** 4
**Contribution:** 3
**Rating:** 8
**Confidence:** 3

**Summary:**

The paper presents a method for learning more expressive activation functions as products of small neural networks themselves.  Theoretical results are presented on the universal expressivity of this parameterization and its ability to efficiently represent functions which have low rank structure to exploit.  Principled initialization schemes are proposed and theoretically justified.  Experiments on function reconstruction tasks show a significant improvement compared to state of the art architectures, while experiments with standard classification tasks show a marginal improvement to test accuracy.

**Strengths:**

The paper is very well written with a clear exposition and convincing experimental results.  The theoretical results were also presented in a clear and complete manner.  I particularly appreciated the commentary and theoretical results on parameter complexity, this is often omitted during discussions of universal approximation.  This work gives a significant improvement to architectures for signal reconstruction specifically and should be accepted.

**Weaknesses:**

Some minor comments for improving the development:

The commentary on parameter sharing at the end of Section 3.2 was of immediate interest during a first read but is minimal in the main text and pushed to the appendix.  An extra comment here and at least a reference to where this appears in the appendix would make for a smoother read.

There is a single comment on memory considerations in the concluding remarks, but having more commentary on this would be very useful for practitioners interested in using this method.

**Questions:**

1. What are the architecture size/shape parameters (depth/width) of the small MLPs used for the g's throughout?

2. How sensitive is the performance to these architecture size choices for the g's?  As I understand the scaling law experiments increase the parameter count with the width of the small MLPs, is there sensitivity to the number of layers or rank as well?

3. Is there a typo in eq (2) and where this appears in the appendix as well?  An extra comma on the right-hand side.

---

> ### Author Response · Authors · 2025-11-27
>
> Thank you for your careful assessment. We appreciate in particular that you found both the exposition and the parameter-complexity analysis useful. We have uploaded a revised manuscript that incorporates your suggestions to improve clarity and practical guidance.
>
> ## 1. Parameter Sharing
> > "The commentary on parameter sharing at the end of Section 3.2 was of immediate interest during a first read but is minimal in the main text and pushed to the appendix. An extra comment here and at least a reference to where this appears in the appendix would make for a smoother read."
>
> In Section 3.2, we have expanded the parameter sharing discussion to more clearly contrast the flexible LRNN (distinct product-structured activations per neuron) with the shared-activation variant, where the univariate component functions are shared across neurons within a layer.
>
> The main text now explicitly points to the corresponding ablation study in Appendix C.1 (Figure 11), which compares the flexible LRNN-SPDER (“Flex”) with the shared-activation (“SA”) variant on image representation tasks. This study shows that parameter sharing can improve parameter efficiency at smaller model sizes, while distinct activations are important to maximize fidelity for complex, high-frequency signals.
>
> ## 2. Memory Considerations
> > "Having more commentary on [memory] would be very useful for practitioners..."
>
> We agree. This is an important practical consideration. We have added a detailed memory analysis in **Appendix B.2: Memory Complexity Analysis and Optimization Strategies**.
>
> The concluding section now explicitly highlights Appendix B.2 as the place where these implementation strategies are detailed.
>
> In Appendix B.2, we derive the memory scaling with respect to projection width $\bar{d}$, rank
> $r$, batch size, and component-MLP width. We identify the backward pass of the product term as the primary bottleneck.
> We outline specific mitigation strategies such as gradient checkpointing (which removes the memory dependence on the component MLP width) and kernel fusion (which avoids materializing the large intermediate tensor by performing projection, component evaluation, and product reduction within a fused kernel). We also note that Lemma 1 (variance-controlled initialization result) indicates that LRNNs are well-suited for mixed precision training.
>
> ## 3. Architecture and Sensitivity of $g$
> > “What are the architecture size/shape parameters (depth/width) of the small MLPs used for the g's throughout?”
> “How sensitive is the performance to these architecture size choices for the g's? As I understand the scaling law experiments increase the parameter count with the width of the small MLPs, is there sensitivity to the number of layers or rank as well?”
>
> Each univariate component function $g_j$ is implemented as a small, shallow MLP (Section 3.2). In all our experiments, they are shallow MLPs with a single hidden layer containing only 1 to 4 hidden neurons. We increase the rank (number of LRNN neurons) when scaling the model and keep the architecture of the shallow MLPs for the component functions fixed as follows:
>
>  - **Image representation:** MLPs with one neuron in the hidden layer (SPDER activation), chosen so that the LRNN parameter count matches SIREN/SPDER baselines (Table 5 in Appendix E).
>  - **Audio representation:** MLPs with four neurons in the hidden layer (SPDER activation); a three-layer LRNN with rank 118 and $\bar{d}=10$ is used to match five-layer baselines (Appendix F).
>  - **PDE benchmark:** MLPs with one neuron in the hidden layer (SIREN activation), with ranks varied in {$\{16,32,48,64\}$} to study scaling with rank (Appendix G).
> - **CT reconstruction:** MLPs with four neurons in the hidden layer (SPDER activation) to match the complexity of the baseline architectures (Appendix H).
>
> We did not observe meaningful gains from increasing the depth of $g_j$. Instead, our empirical findings indicate that, for a fixed parameter budget, it is more effective to allocate capacity to increasing the *separation rank* $r$, and
> the *projection width* $\bar{d}$, rather than making each $g_j$ deeper or wider. Overall, our experiments support the design principle that LRNNs achieve expressivity primarily through *interactions of many simple components* rather than through complex univariate factors. In the revised paper, we include a large-scale study on 1,000 images (Section 4 and Appendix E.1) that provides additional empirical evidence supporting this.
>
> ## 4. Typos
> > "Is there a typo in eq (2) and where this appears in the appendix as well? An extra comma on the right-hand side."
>
> - Thank you. We have removed the extraneous comma in Equation (2) and corrected the corresponding expression in the appendix proof to maintain consistency.
> ---
> We hope these clarifications and revisions make the paper more useful for both theoreticians and practitioners. We thank you again for your constructive feedback.

---

### Author Response · Authors · 2025-12-04
**Summary of contributions and revisions**

Dear Area Chair(s),

We thank the reviewers for their feedback, which enabled us to strengthen the theoretical foundations and empirical scope of our work. We summarize below the core contributions and revisions.
## Summary of contributions
We introduce *low-rank separated neural networks (LRNNs)*, a deep learning architecture based on a generalization of the classical notion of low-rank function decomposition. Each LRNN neuron is equipped with a learnable product-structured activation:
$$\phi(z)=\prod_{j=1}^{\bar{d}}\ \bigl( 1 + \gamma\ g_j(z_j)  \bigr),$$
where  $z=Wx+b \in \mathbb{R}^{\bar{d}}$ and $g_j: \mathbb{R} \rightarrow \mathbb{R}$ is a learnable univariate component function.

In contrast to standard MLPs where all neurons share a fixed scalar activation, each LRNN neuron learns a **distinct** factorized activation operating on a $\bar{d}$-dimensional projection. LRNNs recover MLPs as a special case when $\bar{d}=1$ and $g_j$ is a standard activation.

We present a detailed theoretical analysis establishing that LRNNs retain the universal approximation guarantees of MLPs (Theorem 1) in addition to the following properties:

1. **Variance-controlled initialization (Lemma 1):** The scaling $\gamma = \bar{d}^{-1/2}$ ensures that both activation variance and cumulative gradient variance remain $O(1)$ independent of projection width, supporting stable training of wide product structures.

2. **Curse-of-dimensionality mitigation (Theorem 2):** For $d$-dimensional functions whose functional ANOVA decomposition is dominated by low-order terms (i.e., low effective superposition dimension), LRNNs achieve $\varepsilon$-approximation with parameter complexity $O(\mathrm{poly}(d)/\varepsilon)$ rather than exponential dependence on $d$. This formalizes when and why LRNNs are efficient.

3. **Combinatorial frequency synthesis (Lemma 2):** A single LRNN neuron equipped with $\bar{d}$ periodic component functions generates $2^{\bar{d}}-1$ distinct frequency combinations, versus a single frequency pair per neuron for a SPDER/SIREN MLP. This explains the superior spectral fidelity of LRNNs on image and audio benchmarks.

In summary, our theoretical analysis provides crucial insights into when and why the distinct inductive bias of product-structured activations leads to expressive and parameter-efficient architectures.
## Empirical results
To address the comments by *Reviewers XNN7* and *VXpv* on evaluation scale and time-to-solution, we conducted a **large-scale study on 1,000 ImageNet images** (3 seeds each, 3,000 models per architecture) measuring success rate and wall-clock time to reach target PSNR thresholds at comparable parameter budgets (~200k):

|Target PSNR|LRNN-SPDER|SPDER|SIREN|
|-------------|------------|-------|-------|
| 40 dB| **100%** success (7.0s avg)| 26.4% success| 1.8% success|
| 35 dB| **100%** success (4.9s avg)| 83.8% success| 26.5% success|

At the 40 dB threshold, LRNN-SPDER succeeds on every image, whereas SIREN and SPDER fail in 98.2% and 73.6% of cases, respectively, demonstrating robustness across diverse natural images.

Beyond images, we demonstrate consistent gains across three additional domains:
- **Audio:** **3–11x** lower MSE with *superior spectral fidelity* across speech/instrumental/vocal signals
- **PDEs:** **8x** parameter reduction vs. SIREN, up to orders of magnitude lower error than KANs
- **Sparse-view CT:** Near artifact-free reconstruction in the limited projections regime (50-100), where baselines exhibit visible degradation.
## Summary of Revisions
- **Section 4 & Appendix E.1:** New 1,000-image ImageNet study with success rates and time-to-solution (addresses Reviewers VXpv, XNN7)
- **Appendix A.6:** Detailed discussion of Theorem 2 assumptions with examples of function classes that satisfy/violate them (addresses Reviewer yobX)
- **Appendix B.2:** Memory complexity analysis and implementation aspects such as checkpointing, kernel fusion, mixed-precision (addresses Reviewers XhCn, VXpv)
- **Appendix A.2:** Interpretability via component function visualization and interaction heatmaps (addresses Reviewer VXpv)
- **Section 3.2:** Expanded parameter-sharing discussion with pointer to ablation study in Appendix C.1 (addresses Reviewer XhCn)
- **L74–81:** Clarified that we leverage low-rank *function* decomposition to model higher-order interactions, distinct from low-rank model compression (addresses Reviewer XNN7)
- **Other changes:** Updated **Section 4** to summarize ablation studies, updated figures for improved legibility.

##
The reviewers' constructive feedback enabled us to provide a large-scale empirical demonstration of robustness, and a clearer exposition of the theoretical foundations connecting signal structure to architectural design. We believe the revised paper presents LRNNs as a principled, versatile building block with both strong theoretical grounding and demonstrated practical utility across multiple domains.

Sincerely,

Authors

---

### Meta-Review · Area_Chair_fTSW · 2026-01-07

**Summary:**

Reviewers’ concerns can be categorized into three main points:
- Empirical rigor and scale: Initial versions were criticized for limited evaluation breadth (few images/signals, unclear robustness across datasets) and lack of large-scale or statistically meaningful studies.
- Practicality: Reviewers questioned the computational and memory overhead of product-structured activations, requesting clearer comparisons on wall-clock time, memory scaling, and feasibility for large-scale use.
- Theoretical clarity: Some reviewers found the assumptions and implications of the low-rank/ANOVA-based theory (especially Theorem 2) insufficiently intuitive, and asked for clearer explanations of when the theory applies and what kinds of functions violate it.

During the rebuttal period, the concerns have been addressed quite comprehensively. Thus, I recommend acceptance.

**Reviewer Concerns:**

All concerns have been addressed well.

**Reviewer Scores:**

Reviewer XNN7 was initially negative, but as it seems that the concerns have been addressed carefully, it is highly probable that the reviewer may have raised the score.

---

### Decision · Program_Chairs · 2026-01-26

Accept (Poster)